# The Latest Advances in Ink-Based Nanogenerators: From Materials to Applications

**DOI:** 10.3390/ijms25116152

**Published:** 2024-06-03

**Authors:** Bingqian Shao, Zhitao Chen, Hengzhe Su, Shuzhe Peng, Mingxin Song

**Affiliations:** 1School of Applied Science and Technology, Hainan University, Haikou 570228, China; 181233@hainanu.edu.cn (B.S.); 20213005490@hainanu.edu.cn (Z.C.); 20213005591@hainanu.edu.cn (H.S.); 20213005551@hainanu.edu.cn (S.P.); 2School of Electronic Science and Technology, Hainan University, Haikou 570228, China

**Keywords:** energy harvesting, ink-based nanogenerators, optimization of ink materials, device structural design, printing technology

## Abstract

Nanogenerators possess the capability to harvest faint energy from the environment. Among them, thermoelectric (TE), triboelectric, piezoelectric (PE), and moisture-enabled nanogenerators represent promising approaches to micro–nano energy collection. These nanogenerators have seen considerable progress in material optimization and structural design. Printing technology has facilitated the large-scale manufacturing of nanogenerators. Although inks can be compatible with most traditional functional materials, this inevitably leads to a decrease in the electrical performance of the materials, necessitating control over the rheological properties of the inks. Furthermore, printing technology offers increased structural design flexibility. This review provides a comprehensive framework for ink-based nanogenerators, encompassing ink material optimization and device structural design, including improvements in ink performance, control of rheological properties, and efficient energy harvesting structures. Additionally, it highlights ink-based nanogenerators that incorporate textile technology and hybrid energy technologies, reviewing their latest advancements in energy collection and self-powered sensing. The discussion also addresses the main challenges faced and future directions for development.

## 1. Introduction

With the advent of the era of the Internet of Things (IoT) and intelligent technologies, wearable flexible electronic devices have experienced rapid development, with self-powered flexible energy sources revealing substantial potential for application [1]. The design of nanogenerators is relatively straightforward, capable of harvesting energy from the surrounding environment and achieving efficient energy conversion, which has sparked significant research interest [2]. Furthermore, the energy sector is currently facing the dual challenges of an energy crisis and the daunting task of developing renewable energy sources [3]. Studies have shown that the recovery of waste heat generated by industrial and transportation activities could potentially be converted into 15 terawatts of electrical power [4]. Additionally, mechanical energy is ubiquitously present in the environment in various forms, such as sound waves, joint movements, and the rolling motion of vehicle tires [5]. Nanogenerators offer an excellent solution for sustainable energy, as they are capable of recovering widely available renewable energies like thermal and mechanical energies that are present in our daily lives. Through the relentless efforts of researchers, nanogenerators have made continuous breakthroughs in wearable power technology, with increasing peak output power density, a growing variety of types, and enriched manufacturing techniques. Nanogenerators also show broad application prospects in areas such as motion tracking, physiological monitoring, intelligent humanoid robots, and human–machine interactions [6,7,8,9]. Compared to traditional rigid batteries, which have limitations in size reduction, finite usage cycles, and significant environmental pollution risks, and are incapable of harvesting the dispersed, low-energy density, and low-frequency energies from the surrounding environment, nanogenerators represent an exciting future direction for portable power sources or self-powered systems in wearable electronics [10].

The environment around us abounds with green and sustainable micro–nano energy, epitomized by ambient light, waste heat, and mechanical forces. These forms of energy are ubiquitous, yet traditional generators are inept at efficiently capturing and utilizing them. The concept of the nanogenerator, introduced by Professor Zhong Lin Wang and his team [11] in 2006, forged a novel avenue for the harvesting of mechanical energy. This breakthrough has since thrust nanogenerators into the limelight, with particular emphasis on triboelectric nanogenerators (TENGs), thermoelectric nanogenerators (TEGs), and piezoelectric nanogenerators (PENGs) [12,13]. In the quest to perpetually enhance energy conversion efficiency, traditional manufacturing methods such as chemical vapor deposition struggle to meet the requirements for precise control over material structure and properties. As a result, additive manufacturing (AM) technologies, notably three-dimensional (3D) printing, have progressively entered the fray, offering new possibilities for the fabrication of these advanced materials [14,15].

AM technologies for nanogenerators, including inkjet printing, screen printing, rod coating, roll-to-roll printing, and 3D printing, facilitate the production of what are termed ink-based nanogenerators. Furthermore, the structural design of nanogenerators plays a pivotal role in enhancing their performance. Leveraging computer-aided design [16], AM offers increased flexibility in the structural design of nanogenerators. The inks used in AM are compatible with a diverse array of materials such as metals, polymers, ceramics, and carbon-based substances, providing a wide range of material sources [17]. Therefore, research into functional inks for ink-based nanogenerators has emerged as a significant area of interest. The innovations driven by ink-based nanogenerators offer an exciting development pathway within the domain of flexible electronic nanogenerators.

In recent years, the development of printed nanogenerator devices has seen significant advancement, particularly ink-based nanogenerators, which have garnered substantial interest from the research community for their innovative functional ink designs and device configurations. As illustrated in Figure 1, in this review, we articulate the latest advancements in nanogenerator ink materials, structural design, and more, through a comparatively parallel framework of ink-based TEG, ink-based TENG, and other ink-based nanogenerators (including moisture-enabled, piezoelectric, and hybrid nanogenerators). The discussion on ink materials embarks from an analysis of the synthesis and doping processes of nanogenerator inks, considering a variety of substrate materials such as metals, polymer composites, and carbon-based materials. Furthermore, in the section on structural design, starting from planar and 3D structures, we discuss the latest advancements in performance improvement through structural enhancements of ink-based nanogenerators. To conclude, this paper presents an overview of the most recent application trends of ink-based nanogenerators, with the intention of inspiring future research in the realms of novel ink materials and advanced structural designs, thereby promoting progress in the field.

## 2. Ink-Based TEGs

In comparison to conventional fabrication methods, such as chemical vapor deposition, magnetron sputtering, and co-evaporation, TEGs developed using printing technologies are distinguished by their significantly lower costs and shorter production cycles. The inception of the first TEG fabricated through inkjet printing was documented in 2014 [48]. The advancement in TEG performance has been predominantly concentrated on the development of TE inks and the optimization of heat collection designs. This paper provides a review of the recent progress in ink-based TEGs, addressing developments in TE inks, heat collection strategies, and other relevant areas, including substrate protection, enhancement of TEG flexibility, and the introduction of novel fabrication techniques. The performance of different thermoelectric devices can be compared using the ZT value (thermoelectric figure of merit), which is defined as [4]:(1)$$ZT=\frac{S^{2}}{\kappa R}T$$
where *S* is the total Seebeck coefficient of the device, κ is the thermal conductivity, *R* is the electrical resistance, and *T* is the average operating temperature of the device. Optimizing this factor satisfies the condition for maximum power output, thus serving as a comparative metric for thermoelectric devices. Additionally, this review extensively uses the following four metrics to compare the performance of different nanogenerators: maximum output power (P_*max*_), maximum output power density, open-circuit voltage (V_*OC*_), and short-circuit current (I_*SC*_).

### 2.1. Optimization of TEG Ink Materials

TE materials possess the ability to directly transform thermal energy into electrical energy [49], with the efficiency of this conversion being directly influenced by the material’s Seebeck coefficient, electrical conductivity (σ), and thermal conductivity [50]. Present research is predominantly centered on metal nanomaterials [51], polymer composites [52], and carbon-based materials [53,54,55], each presenting unique advantages. Flexible TEG devices are progressing towards high performance characterized by affordability, flexibility, and scalability. Bulk binary chalcogenides, particularly bismuth telluride (Bi_2_Te_3_), antimony telluride (Sb_2_Te_3_), and their alloys, exhibit the highest ZTs at ambient temperatures [56]. Alloys based on Bi_2_Te_3_ have achieved a ZT of 1.96 at 420 K, while tin selenide (SnSe) single crystals have demonstrated unprecedented ZTs in the medium-to-high temperature range of 450 K to 850 K [57]. Conversely, polymer composite TE materials are acknowledged for their enhanced printability and suitability for flexible wearable devices, typically characterized by high σ and low κ, yet their TE performance is often limited by a low S [58,59]. Meanwhile, carbon-based nanomaterials, such as carbon nanotubes (CNTs), graphene, carbon quantum dots, and conductive carbon black [25,60], have garnered extensive attention due to their non-toxicity and superior electrical and mechanical properties. Semiconductor colloidal quantum dots (CQDs) have been recognized for their exceptional performance in the TE and optoelectronic domains, with their room-temperature TE performance nearing that of bulk Bi_2_Te_3_-based systems. These CQDs can be synthesized in bulk via straightforward wet chemical methods [61]. Numerous reviews have provided comprehensive summaries of the advancements in these four principal categories of TE materials from various perspectives. For instance, Tian-Ran Wei et al. [62] have provided an overview of the latest research on Ag_2_Q-based (Q = S, Se, Te) silver chalcogenide TE materials, highlighting the interrelation between “composition-phase structure-thermoelectric/mechanical properties”. Longhui Deng et al. [63] explored the structural and performance distinctions of high-performance polymer TE materials from the perspectives of molecular engineering and doping modulation for polymer TE materials, organic small molecule TE materials, and OTE materials. Mengxia Liu et al. [64] revisited the evolution of CQD electronics, particularly emphasizing the latest developments in TE devices. As one of the current focal points of research [65], fiber-based TE materials have been thoroughly reviewed by Wen-Yi Chen et al., who summarized the latest progress in fiber-based TE materials derived from inorganic, organic, and hybrid materials and examined the newest studies on TE devices with 1D, 2D, and 3D structures. Yanan Shen et al. [66] discussed the strategies for enhancing the performance of fiber-based TE materials, fiber processing technologies, and their extensive application prospects. This section will specifically review the recent advancements in ink materials for ink-based TEGs from four perspectives: metal nanomaterial-based TE inks, polymer-composite-based TE inks, carbon-based TE inks, and CQD-based TE inks. In comparison, polymer-composite-based TE materials are preferred due to their outstanding printability and compatibility with flexible wearable devices. Within this category, conductive polymers display high σ and low κ, yet their TE performance is typically hindered by a low S [67].

#### 2.1.1. Metal Nanoparticle-Based TE Inks

Two-dimensional transition metal dichalcogenides (2D TMDs) crystalline nanomaterials, characterized by their atomic-level thickness and large surface area crystal facets, possess distinctive physical and chemical attributes [68,69]. Presently, TE materials based on metal nanomaterials have undergone extensive investigation, resulting in ink-based TEGs that exhibit high-quality power output. Nonetheless, to overcome challenges associated with flexibility and biocompatibility in TEGs, it is imperative to acknowledge the intrinsic limitations of these materials. Notably, rare earth elements such as tellurium (Te) and bismuth (Bi) are not only costly but may also present potential toxic risks. Furthermore, the natural rigidity of these materials hampers their utility in augmenting the flexibility of TEG devices [70]. Several reviews have comprehensively examined the current state and advancements in various inorganic TE materials from diverse viewpoints. For example, Delong Li et al. [71] have explored the capabilities of 2D TE materials like graphene, black phosphorus, TMDs, IVA-VIA compounds, and MXenes, particularly highlighting their novel applications in photodetection integration. Metal halide perovskites have also demonstrated significant promise in TE applications. Sile Hu et al. [72] have outlined the benefits of organic–inorganic (hybrid) halide perovskites and their low-dimensional counterparts, discussing potential approaches to enhance the ZT values of these materials. Recently, Jinfeng Dong et al. [73] provided an overview of the latest progress and methods for performance enhancement in TE devices constructed from nanowires (NWs), thin films, and nanocrystals.

In the fabrication of water-based printing inks, the liquid-phase production process is crucial, predominantly utilizing two main methods: mechanical exfoliation based on tensile/shear stresses and chemical exfoliation facilitated by alkali metal intercalation. The latter is particularly esteemed due to its efficacy in achieving yields near 100% [74]. Additionally, the TE properties of 2D TMDs work well at room temperatures. This has sparked much research. The goal is to create 2D TMD inks using chemical methods in solutions. This method enables the synthesis of 2D TMD materials with diverse nanostructures, such as nanoparticles (NPs), NWs, nanotubes (NTs), and nanoflowers, by incorporating auxiliary techniques including chemical vapor deposition [75], ultrasonic assistance, chemical reduction, and electrocatalysis [76]. Such versatility in synthesis approaches offers multiple strategies for reducing the κ of TE materials while also ensuring that the resultant metal nanomaterials possess both processability and electrical performance. However, realizing high-performance TEGs with solution-processed materials poses a significant challenge. Prior research on solution-processed PbTe-based metal nanomaterials reported a maximal ZT value of merely 0.6 [77]. Conversely, in a more recent study, Yu Zhang et al. [57] demonstrated the preparation of a molecular precursor solution by dissolving Sn and Se ions, followed by the decomposition of these precursors to yield 2D SnSe nanosheets, as depicted in Figure 2a. The subsequent hot pressing process yielded bulk SnSe nanomaterials and printed layers. It was observed that the incorporation of a trace amount of Te during the precursor preparation stage significantly enhanced the TE performance in the direction perpendicular to the substrate layers. Moreover, the environmentally benign 2D TMD material Cu_2_ZnSnS_4_ (CZTS) can be converted into a semiconductor with an adjustable bandgap, either CZTSe or CZTSSe, through partial substitution with Se. Additionally, Cu_2_SnS_3_ (CTS) is recognized as an exemplary p-type semiconductor material. Ubaidah Syafiq et al. [78] developed a series of CZTS, CZTSe, CZTSSe, and CTS inks via hot injection synthesis and ball milling powder synthesis, employing these inks to fabricate p-type TE films. By coupling these with n-type AZO films, they engineered four variants of high-performance TEGs, as shown in Figure 2b. Ethylenediamine has been identified as capable of facilitating charge transfer by removing insulating organic ligands from the nanocrystal surfaces. Most recently, Defang Ding et al. [56] reported an enhancement in the electrical performance of a solution-processed Bi_2_Te_3_ nanosheet ink mixture through the incorporation of ethylenediamine and reduced graphene oxide (rGO) nanosheets, with Figure 2c illustrating the preparation methodology of this ink mixture.

Creating high-performance 2D TMDs inks involves solution-based chemical methods, and it shows great potential, offering broad industrial applications. [79,80]. Nevertheless, the scalability of ink-based TEGs critically depends on the inks’ long-term storage capabilities, necessitating enhanced stability and dispersibility. It is crucial to prevent degradation, agglomeration, and precipitation of the inks during the printing process, as these phenomena directly impair the TE performance of ink-based TEGs [81]. Contemporary large-scale solution chemical synthesis predominantly relies on chemical exfoliation techniques facilitated by alkali metal intercalation [82,83]. However, selenide or telluride-based 2D TMD monolayers, produced via this method, are more susceptible to oxidation over prolonged storage periods due to their distinct electronic properties, compared to their sulfide-based counterparts. This susceptibility leads to ink degradation and TMD monolayer aggregation, thus undermining the inks’ stability and dispersibility [84], which has sparked considerable research interest [79,83,85]. Research conducted by Hyemin Park et al. [74] revealed that oxygen dissolved in aqueous solutions could readily oxidize niobium, forming niobium oxide and causing charge loss. This oxidation also precipitates surface Se NPs, forming thicker NbSe_2_ flakes, adversely affecting the ink’s stability and dispersibility. By incorporating the eco-friendly antioxidant L-ascorbic acid (L-AA), which engages in redox reactions with dissolved oxygen to produce electrons, they formulated NSs-based TE inks with commendable dispersibility and extended stability (as depicted in Figure 3a), maintaining over 87% of the original ZT value even after 30 days of storage. Although the inclusion of surfactants or secondary solvents in the ink formulations can improve dispersibility and stability, these additives are challenging to remove, potentially diminishing the TE performance of the end products. Jingjie Du et al. [81] synthesized various types of one-dimensional metal chalcogenide NWs (e.g., Ag_2_Te, Cu_7_Te_4_, and Bi_2_Te_2.7_Se_0.3_) (illustrated in Figure 3b) through a directional chemical transformation process utilizing Te as a template. This innovation led to the development of high-performance, additive-free stable TE inks suitable for inkjet printing, showcasing the method’s broad applicability across diverse material systems.

One of the prevailing challenges in the fabrication of screen-printed TE films lies in achieving satisfactory TE performance, with a particular emphasis on electrical transport properties. Printed devices incorporate metal nanomaterials for superior electrical transport. This requires robust interfacial connections between the NPs [89]. Inadequate interfacial bonding between NPs leads to diminished carrier mobility. This is a significant barrier to the performance of printed devices [90]. The employment of organic binders in the sintering process of TE inks leads to the formation of porous structures, which in turn create extended pathways for electron transport, thus contributing to lower electrical conductivity. Prior research has indicated that inorganic nano-solders can enhance grain crystallization and interface bonding, thereby effectively improving the electrical conductivity of TE films [91]. As such, the use of nano-solders to facilitate the development of conductive channels during the sintering of TE films has been identified as an efficacious approach to augment the performance of screen-printed TE films [92,93]. As depicted in Figure 3c, Tony Varghese et al. [86] employed a Te-based nano-solder technique to bridge the interfaces between ball-milled Bi_0.4_Sb_1.6_Te_3_ NPs. This approach yielded flexible p-type BiSbTe films, post-sintering, with an exceptionally high room-temperature power factor of 3 mW m^−1^ K^−2^. In addition to the employment of nano-solders for enhancing electrical transport, strategies involving the direct bridging of NPs have also attracted research attention. Figure 3d illustrates the n-type TE thick films produced by Soo-ho Jung et al. [87], which were fabricated using a slurry composed of edge-oxidized graphene doped and dispersed copper-doped Bi_2.0_Te_2.7_Se_0.3_ (BTS). The incorporation of edge-oxidized graphene served to bridge the dispersed BTS particles and defect-free graphene layers within the porous TE thick films, establishing a rapid conductive pathway. This innovation not only bolstered the electrical conductivity but also enhanced the Seebeck coefficient of the TE film. The single n-type device fabricated with this methodology, when subjected to a temperature differential of 80 K, demonstrated an output power that was five times that of the untreated BTS films. Moreover, aerogels, which are nanoscale porous solid materials, are known for their exceptional thermal insulation properties due to their distinctive pore structures. The incorporation of CNTs has been shown to significantly ameliorate the electrical conductivity of aerogels [94]. K. Zhao et al. [88] introduced a technique that involves doping p-type and n-type Bi_2_Te_3_ TE materials with CNTs and aerogels, leading to the successful printing of high-performance TE films (Figure 3e). Notably, traditional TEG fabrication methods often entail prolonged high-temperature sintering and curing processes aimed at enhancing the interfacial connections and grain boundaries of TE particles, which are markedly energy-intensive. Eunhwa Jang and collaborators [95] have successfully established efficient conductive pathways by leveraging a heterogeneous distribution of commercially available p-type Bi_0.5_Sb_1.5_Te_3_ (BST) particles of varying sizes, utilizing smaller particles to fill the interspaces between larger ones [95].

High-performance TE nanogenerators based on ink formulations not only require the inks to possess superior electrical properties but also necessitate specific rheological characteristics due to their unique printing processes. For instance, extrusion-based direct ink writing (DIW) processes demand inks with viscoelastic properties within a specific range [4,96], which is crucial for ensuring precise ink flow through the nozzle and the structural integrity of the printed constructs [97]. Overall, appropriate rheological and mechanical properties are instrumental in enhancing stability, facilitating extrusion, and improving resistance to the deformation of printed shapes during the DIW process [98]. Viscous printing pastes for TE materials can be prepared by adding high-viscosity organic substances such as glycerol and ethylene glycol. However, the residual presence of insulating organic compounds post-sintering can alter the ink composition, leading to a decrease in the electrical conductivity of the printed materials and, consequently, diminished film TE performance. Bo Wu et al. [99] utilized hydrogen bonding between carboxylated cellulose nanofibers (CCNs) to form a physically cross-linked and entangled hydrogel network that restricts the flow of 1D nanorod (NR) dispersions such as Cu${\;\;}_{y}$Te and Ag${\;\;}_{x}$Te NRs, thereby developing high-viscosity p-type and n-type hydrogel pastes (Figure 4a,b). Zhengshang Wang et al. [23] reported a high-phase-stability Bi${\;\;}_{2}$Te${\;\;}_{3}$-based ink that allows direct printing and adjustable structures, as shown in Figure 4c. This was achieved by inducing adsorption layers on the TE particle surfaces through polyelectrolyte additives such as poly(acrylic acid) (PAA) and poly(ethyleneimine) (PEI), improving the ink’s stability and viscoelasticity. The addition of methylcellulose (MC) as an additive enhanced the yield stress and structural recovery, thereby improving the strength properties. Employing charge control and framework reinforcement strategies, materials with high figure-of-merit ZT values of 0.65 for p-type and 0.53 for n-type were obtained. Furthermore, the research team used low concentrations of organic binders PAA and PEI to form stable hybrid structures through electrostatic interactions and steric hindrance, thus dispersing Bi${\;\;}_{2}$Te${\;\;}_{3}$-based TE particles, as illustrated in Figure 4d. Concurrently, a second-phase interfacial welding technique was applied, using a liquefied phase to fill the interparticle voids, enhancing the electrical interconnections between particles [100]. Eunhwa Jang and colleagues [95] combined commercially available p-type BST particles, natural n-type Bi particles, and chitosan binder to formulate a TE composite ink. This chitosan binder achieved effective adhesion at low weight percentages while minimally impacting the film’s electrical conductivity.

The addition of organic binders to inks is a common strategy to easily achieve the desired viscoelastic properties [101]. However, these binders cannot be fully removed during subsequent sintering processes, potentially acting as contaminants that compromise the electrical performance of printed constructs. Moreover, organic binders may result in inadequate electrical interconnections among metal NPs, interrupting the continuity of charge carriers at the interfaces of the printed film particles, thus adversely affecting the TE performance of the films [87]. Prior research has indicated that the viscoelasticity of TE colloidal inks can be modulated through the adjustment of inorganic chemical ions [102]. While increasing the concentration of metal NPs can enhance viscoelastic properties, it may also lead to the clogging of nozzles in microelectronics printers. Fredrick Kim and colleagues [96] have developed high-viscoelastic p-type and n-type all-inorganic TE inks based on (Bi,Sb)${\;\;}_{2}$(Te,Se)${\;\;}_{3}$ by optimizing particle size, distribution, and surface oxidation. Smaller average particle sizes and narrower particle size distributions can yield colloidal inks with higher dynamic viscosity ($\eta^{\prime }$). Controlled surface oxidation, achieved by introducing chalcogenometalates (ChaM), not only improved ink stability but also circumvented the potential shielding effect of the surface electrostatic charges of TE particles by ethylenediamine counter ions in ChaM anions, preventing ink aggregation and ensuring good dispersibility (Figure 5a). Seungjun Choo and his team [103] have developed a Cu${\;\;}_{2}$Se-based 3D printing ink with ideal viscoelastic and customizable rheological properties, devoid of organic binders, utilizing the electrorheological effect of Se${\;\;}_{8}^{2-}$ poly-anions (Figure 5b). Furthermore, Se${\;\;}_{8}^{2-}$ poly-anions served as sintering aids, with the liquefied Se filling the interstitial spaces between Cu${\;\;}_{2}$Se particles through capillary action, facilitating the uniform sintering of Cu${\;\;}_{2}$Se particles at elevated temperatures. Jungsoo Lee [28] and associates proposed an approach to enhance the viscoelasticity of additive-free inks by electrostatically binding Cu${\;\;}_{2}$Se particles, effectively mediated by the surface charge of PbTe particles doped with inorganic Se${\;\;}_{8}^{2-}$ poly-anions from ChaM (Figure 5c). Md Mofasser Mallick and his team [67] reported a novel, scalable one-pot synthesis and preparation method for high-performance Ag${\;\;}_{2}$Se-based n-type printed TE materials. The formation of the orthogonal β-Ag${\;\;}_{2}$Se TE phase through thermal annealing at relatively lower temperatures, due to the dissociation of adsorbed volatile Se by Ag particles, led to high-conductivity transmission paths without defined grain boundaries, achieving a high average ZT of 1.03 (Figure 5d). Additionally, high-entropy alloys, with their disordered and distorted lattice structures, can effectively scatter phonons carrying heat, thereby reducing lattice thermal conductivity and enhancing TE performance. Binbin Jiang et al. [104] achieved strong phonon scattering and a consequent significant reduction in thermal conductivity by introducing alloyed Sn into PbSe materials, forming a cubic phase beyond the solubility limit and thus severely distorting the lattice (Figure 5e). This led to an exceptionally high ZT value of 1.8 at 900 K (Figure 5f).

#### 2.1.2. Polymer-Based TE Inks

Inorganic TE materials based on metal NPs are known for their enhanced stability and superior ZTs [105]. However, these materials frequently depend on scarce metal elements and are characterized by their toxicity, suboptimal mechanical properties, and processing challenges [106]. Consequently, such inorganic semiconductor TE materials fall short of meeting the increasing demands for eco-friendliness and flexibility required by wearable technologies. Conversely, organic materials, particularly conductive polymers, excel in printability but suffer from low κ and high σ, resulting in TE performances that are markedly inferior to those of metal nanoparticle-based counterparts [107,108,109]. Conductive polymers such as poly(3,4-ethylenedioxythiophene):poly(styrenesulfonate) (PEDOT:PSS), polyaniline (PANI), polydimethylsiloxane (PDMS), and silicone rubber are garnering significant interest due to their semimetallic electrical characteristics and their notable advantages of being non-toxic, flexible, and stretchable [110]. The broader application of polymer-based TE materials is challenged by the inherently low S of the materials. The approach of leveraging organic/inorganic hybrid composite materials to integrate the attributes of diverse materials presents a promising avenue to address this challenge [59,111,112,113].

Traditionally, enhancing the low S of conductive polymer TE materials has been achieved through the application of acidic ionic dopants, raising S values to above 1000 S cm${\;\;}^{-1}$ [114,115]. However, this method may adversely affect the materials’ stability and mechanical properties [116]. Solution-processable conjugated polymers (CPs) offer a pathway to circumvent these constraints through small molecule doping [117]. Moreover, the incorporation of metal NPs into conductive organics has been shown to improve the ZT performance of flexible TE materials [111,118], with transition metal doping further enhancing the efficiency of conductive polymer doping [59]. José F. Serrano-Claumarchirant et al. [119] utilized microwave-assisted synthesis to create hexagonal plate-shaped Sb_2_Te_3_ and Bi_2_Te_3_ NPs, embedding them within a polymethylmethacrylate (PMMA) polymer matrix to enhance the interface and connectivity within the conductive network. The employment of dodecanethiol (DDT) as a bridging agent for interface modification increased film compactness, thus improving the electrical conductivity and TE power factor, culminating in the development of p-type and n-type TE inks (as depicted in Figure 6a). The production of long-term stable hybrid films through spin-coating demonstrated the protective role of PMMA towards Sb_2_Te_3_ and Bi_2_Te_3_ NPs. Phillip Won and colleagues [120] explored gallium-based liquid metal (LM) polymers or LM embedded elastomers (LMEE) composites, consisting of a polymer substrate and LM alloys, which exhibit notable stretchability, electrical conductivity, and mechanical robustness. By integrating a highly elastic, conductive, and thermally conducive PDMS silicone rubber polymer substrate and incorporating eutectic gallium-indium (EGaIn) soft-flowing metal through shearing into the precursor, an EGAIN-PDMS emulsion-based LMEE ink was formulated (Figure 6b). Additionally, the application of shock waves encapsulated in high viscoelastic gel disrupted the oxide layer formed by the reaction of EGaIn with oxygen, thereby rendering the DIW electrically insulating LMEE structures conductive. This innovation led to the demonstration of a TEG that showcased enhanced thermal conductivity in the LMEE structure. In the field of green electronic textiles, Sozan Darabi et al. [121] proposed a scalable approach employing PEDOT:PSS-based conductive ink with the addition of ethylene glycol. This ink was applied to regenerated cellulose fibers and yarns spun from ionic liquids using a roll-to-roll coating process, and subsequently augmented with silver NWs to produce machine-washable high-conductivity yarns (Figure 6c). A fully textile-based TEG, fabricated using a conventional sewing machine, achieved a maximum power output (P_*max*_) of 0.2 μW.

#### 2.1.3. Carbon-Based TE Inks

Although polymer TE materials surpass metal nano TE materials in terms of processability, flexibility, affordability, and scalability, their TE performance is relatively inferior. Carbon-based TE materials, such as CNTs and graphene, distinguished by their superior electrical conductivity, mechanical properties, and processability, emerge as viable alternatives to metal nano TE materials and polymer TE materials [122]. Notably, since the discovery of the 2D material graphene in 2004 [71], its exceptional electronic and mechanical properties have attracted significant attention. Additionally, CNTs, encompassing single-walled carbon nanotubes (SWCNTs) and multi-walled carbon nanotubes (MWCNTs), offer distinct advantages [123]. SWCNTs, in comparison to MWCNTs, demonstrate enhanced TE performance and are extensively utilized in the production of flexible TE films. Research on MWCNT-based TE materials is relatively scarce. However, it is important to note that MWCNTs present cost and production benefits [124], and their intricate structure offers improved flexibility, thus providing greater control in optimizing functional designs [106]. CNT-based TE inks are primarily p-type due to the oxidation of CNTs, but n-type CNT materials can be realized through various treatments, such as doping with amine-substituted alkyl dimers, ethylenediamines, or cationic surfactants [125]. Carbon-based TE inks are divided into all-carbon TE inks and carbon composite TE inks. The following sections will delve into these two categories of carbon-based TE inks and explore the latest advancements in carbon-based TE inks integrated with nanofibers (NFs).

The potential of carbon-based composite TE materials has been extensively explored recently, showcasing the ability to amalgamate the advantages of two distinct material types. These composites are characterized by their low cost, low density, low thermal conductivity, and ease of fabrication [126,127]. There exists a strong interdependency among TE parameters, constrained by inherent coupling. The “quantum confinement” concept introduced by L.D. Hicks et al. [128] in the 1990s and the topological insulators theory proposed by Y. Xu et al. in 2014 [129] have provided crucial theoretical foundations for designing nanostructured interfaces in topological insulators to enhance the ZT values of intrinsic materials. Against this backdrop, Bi_2_Te_3_-based alloys, which are quintessential TE materials demonstrating the quantum confinement effect and topological insulators edge states, have garnered extensive attention [130]. Carbon-based nanomaterials, owing to their unique properties, are highly capable of modifying the nanostructures on the surfaces of metal nano TE particles, thereby achieving significantly higher TE performance than intrinsic materials. Moreover, due to their exceptional mechanical properties, these carbon-based materials can enhance the processability and inherent rigidity of the modified metal nanomaterials [131]. At present, n-type Bi_2_Te_3_ TE materials suffer from suboptimal TE performance and stability, which restricts their broader application. Research focused on interface engineering that can effectively improve the ZT values of n-type Bi_2_Te_3_-based TE materials remains insufficient. Peigen Li et al. [130] demonstrated that the incorporation of graphene oxide (GO) can mitigate the pseudo-donor effect in n-type Te-rich environments of Bi${\;\;}_{2}$Te${\;\;}_{3}$-based alloy powders, and the novel interfaces formed with Bi${\;\;}_{2}$Te${\;\;}_{3}$ NPs can enhance phonon scattering. Furthermore, in conjunction with excess Te activation and the liquid phase sintering process, as illustrated in Figure 7a, an improvement in interface connectivity is achieved without an increase in lattice thermal conductivity. In particular, Bi${\;\;}_{2}$Te${\;\;}_{2.5}$Se${\;\;}_{0.5}$ with 1 wt% GO addition reached a peak ZT value of 1.03 at 473 K, with an average ZT of 0.85 in the range of 300–473 K. This also led to a maximum power density of 0.06 W cm${\;\;}^{-2}$ at 154.8 K in a single-pair TE device composed of traditional p-type BST. As illustrated in Figure 7b, inspired by the “cement-rebar” structure, Zhijun Chen et al. [24] utilized SWCNTs as “rebar” and bonded re-crystallized Bi${\;\;}_{2}$Te${\;\;}_{3}$ nanolayers as “cement” onto the surface of SWCNTs, achieving the effective dispersion of Bi${\;\;}_{2}$Te${\;\;}_{3}$ nanolayers in solution. Casting the SWCNTs and Bi${\;\;}_{2}$Te${\;\;}_{3}$ mixture as ink resulted in an n-type Bi${\;\;}_{2}$Te${\;\;}_{3}$/SWCNT hybrid flexible film. This film demonstrated a Seebeck coefficient of −100.00 ± 1.69 μV K${\;\;}^{-1}$ in air and achieved the highest power factor of 517.20 ± 26.21 μW m${\;\;}^{-1}$ K${\;\;}^{-2}$ at 300 K, with an exceptionally low in-plane thermal conductivity of 0.33 W m${\;\;}^{-1}$ K${\;\;}^{-1}$ and a remarkably high ZT of 0.47 ± 0.02. Additionally, the material exhibited both long-term stability and flexibility.

In recent advancements, CNTs have been highlighted for their superior TE properties facilitated through ionic or molecular doping [132,133,134], leveraging their intrinsic carriers of both positive and negative charges along with highly adaptable performance characteristics. Despite the predisposition of pristine CNT TE inks towards p-type behavior due to atmospheric oxygen impurity, n-type attributes can be induced by doping with electron-accepting organic molecules or polymers [135,136,137]. The development of biocompatible, green, water-based CNT inks, rooted in CNT technology, has emerged as a focal point of research. Nevertheless, TE inks based on carbon encounter two predominant challenges: firstly, the energy conversion efficiency of n-type organic TE materials falls short when compared to their p-type counterparts, a gap that solution-processed n-type dopants might bridge; secondly, the propensity for self-aggregation in cost-effective MWCNTs could compromise the inks’ long-term stability without a viable biocompatible solution. Christos K. Mytafides et al. [138] demonstrated effective doping of SWCNTs by incorporating alkylammonium cationic surfactants (CTAB) into aqueous solutions, resulting in the formulation of aqueous CNT inks. Utilizing tandem connections of p-type and n-type SWCNT TE elements, a flexible, metal-free all-carbon organic TEG was constructed, delivering a V${\;\;}_{OC}$ of 1.05 V and a I${\;\;}_{SC}$ of 1.30 mA at a temperature difference of πT = 150 K, with the output power reaching 342 μW. Importantly, traditional chemical and surface modifications, which may compromise CNT surface integrity and thus electrical conductivity, are circumvented by employing green adhesives and dispersants such as positively charged chitin nanocrystal (ChNCs). These ChNCs engage in non-covalent interactions, including π-π stacking, hydrophobic interactions, and electrostatic attractions with negatively charged CNTs, enhancing the dispersion and stability of CNT dispersions, as illustrated in Figure 7c. Utilizing ultrasonication, Yunqing He et al. [25] prepared uniformly dispersed and stable ChNCs/MWCNT (CCNT) inks using ChNCs as both adhesive and dispersant, which maintained 91.1% of their dispersion efficiency and a maximum MWCNT concentration of 33 mg mL${\;\;}^{-1}$ even after three months of storage, as depicted in Figure 7d. The focus on cellulose, a plentiful and eco-friendly material characterized by its low thermal conductivity and density, has intensified due to the derived cellulose nanofibers (CNFs). Their high aspect ratio and propensity to adsorb and entangle CNTs in aqueous environments are particularly noteworthy. Additionally, the interaction between the hydrophobic functional groups on CNT surfaces and the electrostatic repulsion from ionized carboxyl groups among the fibers promotes the uniform dispersion of CNTs in water. By template printing p-type thermoelectric inks based on MWCNTs and carboxylated nanocellulose fibers (C-CNFs) onto a flexible, durable paper substrate, Hongbing Li et al. [106] achieved the creation of the inaugural MWCNT-based thermoelectric paper, as shown in Figure 7e. This TE paper exhibited a 30% improvement in dispersion compared to additive-free MWCNT inks and reached a peak Seebeck coefficient of 12.7 μV K${\;\;}^{-1}$. Ramakrishna Nayak et al. [139] developed a PANI-graphite composite ink, enhancing its printability with cellulose acetate resin and boosting conductivity, crystallinity, and mobility by incorporating dimethyl sulfoxide and graphite into PANI. The microporous structure of the ink films, achieved through screen printing, further contributes to reduced thermal conductivity. These films, along with the insulating properties of cellulose acetate resin, amplify charge carrier scattering, thereby diminishing both the concentration and thermal conductivity of charge carriers.

#### 2.1.4. CQD TE Inks

CQDs have emerged at the forefront of solution-processed semiconductor TE materials, particularly showcasing unique advantages within the domain of flexible thermoelectrics, such as TE sensors and wearable TE devices. Previous research has established that SnSe-CdSe nanocomposite materials achieved an impressive maximum ZT value of 2.2 at 786 K [140]. Semiconductor CQDs not only exhibit TE performance on par with traditional Bi${\;\;}_{2}$Te${\;\;}_{3}$ materials but also demonstrate quantum confinement effects due to their low-dimensional nature and tunable grain boundary density. These effects contribute to an enhanced Seebeck coefficient and reduced thermal transport. In these materials, thermal conduction can be facilitated by both electrons and phonons, thereby allowing the material’s total thermal conductivity to be articulated as the sum of carrier thermal conductivity ($\kappa_{E}$) and lattice thermal conductivity ($\kappa_{L}$), where κ = $\kappa_{E}$ + $\kappa_{L}$. It is noteworthy that the $\kappa_{L}$ of polycrystalline materials correlates directly with the phonons’ mean free path. Integrating quantum dots into bulk TE materials as phonon scattering sites can efficaciously diminish the phonons’ mean free path [64]. Moreover, the adjustable solubility of solution-processed semiconductor CQDs facilitates their compatibility with various low-temperature printing technologies, paving the way for cost-efficient large-scale AM [141]. While CQDs represent a relatively nascent class of semiconductor TE materials, the advancements in this area have been encapsulated in review articles. Mohamad Insan Nugraha et al. [61] have provided a comprehensive overview of the predominant liquid-phase synthesis techniques for semiconductor CQDs, including hot injection, non-hot injection, laser irradiation, aqueous phase synthesis, and microwave-assisted synthesis. They further expounded on the current developments in hybrid CQDs/organic TE semiconductors, accentuating their promising applications in thin-film TEGs. SnSe garners attention for its eco-friendliness, cost-effectiveness, and exemplary TE performance. Despite polycrystalline SnSe not matching the TE efficiency of single-crystal SnSe, its affordability, straightforward production process, and enhanced mechanical properties render it appealing. In polycrystalline SnSe systems, the concurrent realization of high peak ZT values and substantial average ZT values remains a formidable challenge. Shuang Li et al. [142] achieved the synthesis of Ga-doped p-type polycrystalline SnSe utilizing a 5T in situ strong magnetic field-assisted solution synthesis technique. This approach led to the induction of nanopores, quantum dots (Sn, Se), and lattice strains within the dislocation network, culminating in increased carrier concentration, augmented phonon scattering, and decreased lattice thermal conductivity. The resultant 5T-NP/QD Sn${\;\;}_{0.975}$Ga${\;\;}_{0.025}$Se sample exhibited a peak ZT value of 2.0 and a notable average ZT value of 0.74 over the temperature range of 300–873 K, along with a maximum energy conversion efficiency of 12.6%. Furthermore, Mohamad Insan Nugraha et al. [22] enhanced electron transport and preserved low thermal conductivity in lead sulfide (PbS) QDMH-based thin films through chemical doping with Cs${\;\;}_{2}$CO${\;\;}_{3}$ salt. The introduction of Cs${\;\;}_{2}$CO${\;\;}_{3}$ salt as a dopant before the formation of the film not only stabilized the colloidal ink matrix but also amplified the n-type TE behavior of the film at ambient temperature, achieving a significantly lower thermal conductivity compared to existing films.

### 2.2. TEG Structure Design

While numerous researchers have endeavored to advance the power generation capabilities of TEGs from a material science standpoint [49], it is important to recognize that the TE performance of ink-based TEGs does not solely hinge on the ZT values of TE materials. It is also closely intertwined with the structural design of the TEGs [143]. By refining the structural design of TEG systems to enhance thermal transfer, a significant improvement in the TE performance of TEGs can be achieved. The structural design considerations for TEGs encompass the geometric configuration of TEG legs and the design for heat collection [144]. Specifically, factors such as the temperature differential across TEG legs, the spacing and number of legs, as well as their geometric configurations, have a direct influence on the TE performance of TEGs. Even with a constant ZT value for TE materials, augmenting the number of leg pairs can lead to an increase in power output, an effect that remains unaffected by the inter-leg spacing. Nevertheless, to realize TEGs with high energy conversion efficiencies, it is imperative to ensure a substantial temperature differential between legs [145] and to judiciously control the leg spacing. Consequently, the structural design of TEGs plays a pivotal role in their performance. This section will elucidate the specific impacts of various TEG structures—namely planar, 3D, and bulk ink-based TEG structures—on the performance of ink-based TEGs. 

#### 2.2.1. Planar Ink-Based TEG Structure

Planar thin-film TEGs, distinguished by their 2D leg configurations, are amenable to large-scale fabrication via screen printing techniques [90]. This method enables the creation of porous structures within flexible TEGs (FTEGs), where the presence of pores significantly enhances phonon scattering. This enhancement in turn leads to an increase in carrier mobility, a reduction in bulk carrier concentration, and a decrease in lattice thermal conductivity, collectively contributing to an improved TE performance [146]. Thin-film TEGs are particularly valued for their material efficiency and manufacturing convenience [26]. Despite this, the power density output of planar thin-film TEGs is comparatively low, a challenge that can be mitigated through strategic stacking and performance optimization [147]. Specifically, the optimization of 2D TEG leg structures stands out as an effective means of performance enhancement. This section will further discuss the latest advancements in the optimization of planar TEG structures, emphasizing two principal strategies: the structural refinement of 2D TEG legs and the application of stacking lamination techniques.

The primary parameters of 2D TEG leg structures encompass aspects such as leg shape, length, width, and thickness [148]. Extensive simulation studies have substantiated the definitive impact of these structural parameters on TEG performance [149]. It has been established that the length of TEG legs critically influences the internal resistance and temperature gradient within the TEG, which, in turn, indirectly affects the output voltage and power conversion efficiency. The width of TEG legs is associated with thermal stress experienced between the legs, whereas the thickness of the legs pertains to the differential in temperature. Moreover, the shape of the TEG legs directly determines the cross-sectional area of the legs on both the hot and cold sides, significantly impacting thermal efficiency [150]. Through the application of screen printing to fabricate rectangular and trapezoidal PANI/graphite-based TEG legs (as depicted in Figure 8a), Ramakrishna Nayak et al. [151] compared the temperature gradients of the two leg configurations. The findings revealed that the trapezoidal structure yielded a higher temperature gradient, leading to a 2.72-fold enhancement in the Seebeck coefficient and a 3.82-fold increase in power output. Additionally, the utilization of high-viscosity ink in screen printing resulted in thin, porous films, which decreased the thermal conductivity of TEGs by 20.33-fold, augmenting the Seebeck coefficient and power output by 11.53-fold and 8.52-fold, respectively. By employing silver, which surpasses graphene in conductivity, as the contact material, the power output increment was further amplified by 2.17 times. The conclusion was drawn that the optimization of ink viscosity and film porosity exerted a more pronounced effect on the performance enhancement of FTEGs than the leg configuration and material properties. Organic TE materials, such as PEDOT:PSS, have garnered widespread attention due to their elevated ZT values. Muhammad Shakeel et al. [48] opted for a glass substrate instead of a costlier ceramic layer and maintained the temperature gradient between the hot and cold junctions through geometric modifications, as showcased in Figure 8b. Utilizing PEDOT:PSS and silver inks, TEGs with characteristic lengths of 30 and 40 mm were printed, achieving maximum power outputs of 5.17 ± 0.5 nW and 4.08 ± 0.5 nW, respectively. It was observed that an increase in length diminished the overall performance of the TEG, underscoring the future potential for printing transparent TE materials on glass substrates to capture thermal energy. The application of mechanical pressure during the thermal curing process facilitates the production of dense, highly conductive TE films at reduced processing temperatures. Furthermore, the configuration of heat sinks/sources plays a pivotal role in determining TEG performance. Employing screen printing for Bi-Sb-Te (p-type) and Bi-Se-Te (n-type) inks, combined with pressure sintering methods, Pin-Shiuan Chang et al. [26] fabricated a flexible planar TEG capable of delivering a power output of 50 mW under a temperature differential of 54.9 °C. A novel directional heat collection design, incorporating HTL and insulating foam (Figure 8c), was proposed to maximize the heating area of the planar TEG and enhance its performance, achieving a maximum power density of 58.3 mW cm${\;\;}^{-2}$ at a temperature difference of 5.7 °C.

In addition to improvements through planar structures, laminated stacking technology also serves as an effective method for increasing the power output of planar ink-based TEGs. In this approach, the total TE output voltage of the multilayer laminates is the sum of the TE voltages of each individual layer. Doping with single-walled CNTs is considered to optimize TE performance. Functionalized glass fiber (GF) sheets are notable for their low cost, light weight, and ease of processing. Coating with a functional water solution of TE nanocrystals can transform TE-inactive reinforced GFs into TE materials. George Karalis et al. [27] introduced the fabrication of GF-SWCNT fabric layers for the first time by coating SWCNT ink on unidirectional (UD) glass fibers, as illustrated in Figure 8f. The p-type GF-SWCNT fabric layers exhibited a Seebeck coefficient of +23 μV K${\;\;}^{-1}$ and a power factor of 60 μW m${\;\;}^{-1}$ K${\;\;}^{-2}$, while the n-type layers displayed a Seebeck coefficient of −29 μV K${\;\;}^{-1}$ and a power factor of 118 μW m${\;\;}^{-1}$ K${\;\;}^{-2}$. As shown in Figure 8d, stacking these GF-SWCNT layers to construct a 16-layer structure of glass-fiber-reinforced polymer composite laminate enabled an electrical power output of 2.2 μW under a temperature difference (ΔT) of 100 K. Moreover, by utilizing the interlayer interfaces as TE junctions, with the p-n TE junctions in the overlapping regions and the remainder forming the TE legs, a significant advancement was achieved. Advanced structure FPR composites, combined with nanomaterials, can significantly enhance material properties. Carbon fiber (CF) reinforced materials are favored for their excellent electrical conductivity. Currently, various research groups are dedicated to improving the TE efficiency of large-scale composite materials. George Karalis et al. [152] reported on the preparation of TE paste by mixing different proportions of Te NWs into a PEDOT:PSS matrix, which was then deposited on GF fabric through a coating technique, resulting in TE-functionalized GF layers, as shown in Figure 8g. The TE paste coating with a mass ratio of 1:1 exhibited the best TE performance, with a power factor of 57.2 μW m${\;\;}^{-1}$ K${\;\;}^{-2}$ and an in-plane Seebeck coefficient of +189 μV K${\;\;}^{-1}$. Finally, as demonstrated in Figure 8e, the fabrication of 10-layer UD and cross-laminated laminates, combined with high-conductivity CF bundle electrodes, formed a through-thickness high-performance TEG design. At a ΔT of 100 K, this TEG exhibited a TE voltage of 8.4 mV and a TE current (I${\;\;}_{sc}$) of 597.4 μA. Compared to conventional CFRP UD laminates, a 10% increase in bending strength and modulus was achieved. 

#### 2.2.2. Three-Dimensional Ink-Based TEG Structure

Planar ink-based TEGs harvest temperature gradients within their plane and convert them into electrical signals through TEG legs that adhere closely to the heat source surface. However, a temperature drop at the interface between the TEG and the heat source is inevitable. Additionally, common heat sources in daily life, such as the human body, often exhibit a 3D form, with the primary temperature gradients being perpendicular to their surface. Therefore, TEGs must achieve close and effective thermal contact with 3D heat sources of diverse shapes to prevent thermal losses in air gaps resulting from inadequate contact [4]. The challenge of inefficient thermal contact between TEGs and curved heat sources constitutes a significant obstacle to enhancing the performance of TEGs with low power output [143]. Thus, planar ink-based TEGs confront not only the challenge of low thermal energy collection efficiency but also issues related to limited flexibility and thermal losses [153].

In light of the inherent constraints associated with planar ink-based TEGs, enhancing their TE performance through geometric optimization and strategic heat source allocation becomes imperative [154]. The temperature gradients established in the direction perpendicular to the contact surface can be effectively harnessed by employing TEG legs with 3D configurations, thus endowing TEGs with augmented power outputs [155]. Recent simulation studies have underscored that judiciously designed geometric configurations of 3D TEG legs can substantially elevate the efficiency of energy conversion. Beyond the conventional aspect ratios or cross-sectional comparisons of rectangular TEG legs, a variety of novel geometric designs, such as tubular TEG configurations, are under investigation [156]. Furthermore, to realize enhanced power outputs, 3D ink-based TEGs equipped with 3D legs must ensure flexible conformal contact with 3D heat sources or be seamlessly integrated with thermal systems. Additionally, the design of TEG modules should prioritize compactness, leveraging printing technologies to fabricate dense, expandable arrays of high-performance TEGs, thereby maximizing the output density within constrained spaces [143].

Traditional manufacturing methodologies constrain the precision in the design and customization of module structures, whereas AM offers a path to fabricate components with intricate shapes efficiently and cost-effectively [157]. Conventional 2D TE modules are suboptimal for establishing high temperature gradients, which in turn limits power generation potential. The geometric configuration of TE legs emerges as a pivotal factor influencing the efficiency, durability, and cost-effectiveness of TEGs. DIW-printed 3D TE legs facilitate effective vertical heat transfer, enhancing the system’s TE performance. In waste heat recovery applications, inadequate thermal contact diminishes thermal power output. PbTe-based compounds stand out as among the most suitable for the temperature ranges typical of waste heat emissions. To circumvent compositional alterations during the sintering of organic inks, Jungsoo Lee et al. [28] introduced a tailored design for tubular PbTe legs (Figure 9a), leading to the fabrication of a self-supporting TE tube without the need for a substrate by assembling 3D printed PbTe tubes. Additionally, the anti-corrosive coating technology for waste gases reported by Brostow et al. has extended the operational lifespan of TE materials in corrosive environments. The mechanical durability of module designs, especially under external stresses such as compression or tension, holds practical importance yet has been largely overlooked. Diverging from TE foams composed of oxides, organics, carbon allotropes, and their blends, Seungjun Choo et al. [103] proposed a novel and durable honeycomb TE architecture that surpasses traditional material configurations (Figure 9b). High-power TEGs necessitate 3D TE leg structures with high aspect ratios to achieve substantial temperature gradients. As illustrated in Figure 9c, utilizing 3D printing, Fredrick Kim et al. [96] directly inscribed ink into microscale 3D arch and lattice structures, realizing a high-performance 3D TE architecture. This approach led to the development of microscale TEGs with high-aspect-ratio 3D filaments and TE legs, showcasing a significant temperature gradient and achieving a power density of 479.0 μW cm${\;\;}^{-2}$. Md Mofasser Mallick et al. [158] designed 3D supports in various shapes including rectangular, cylindrical gear, and serrated forms (Figure 9d), subsequently coating them with n-type Ag${\;\;}_{2}$Se TE ink and p-type PEDOT TE ink to construct high-performance 3D structured TEGs. These TEGs delivered a maximum power output of 7 μW at a ΔT of 70 K and a V${\;\;}_{OC}$ of 4.2 mV. To minimize thermal losses through the substrate, M. Massetti et al. [29] introduced a technique that combines direct laser writing to create ablated conical microcavities with inkjet printing to fill TEG legs doped with p-type PEDOT:PSS and n-type fullerene derivatives. This led to the creation of an embedded organic compact TEG architecture (Figure 9e), which demonstrated a power density of 30.5 nW cm${\;\;}^{-2}$ even under minimal thermal gradients. Seongkwon Hwang et al. [30] developed a high-performance viscoelastic TE ink by modulating the concentration of SWCNTs and micro-doping with p-type PAA and n-type PEI. Moreover, as illustrated in Figure 9f, the direct printing of PDMS ultra-thin thermal insulation substrates not only provided arched vertical support but also reduced thermal losses, culminating in the fabrication of high-performance 3D-compatible TEGs. The nozzle-induced stacking effect of CNTs further enhanced electrical conductivity, enabling the TEG to exhibit a Seebeck voltage of 0.28 mV K${\;\;}^{-1}$ cm${\;\;}^{-2}$ and a power density of 1.24 × 10${\;\;}^{-4}$
μW cm${\;\;}^{-2}$.

#### 2.2.3. Bulk Ink-Based TEG Structure

Planar ink-based TEGs, fabricated using thin-film TE materials, struggle to achieve the thickness of traditional vertical TEGs through stacking, thereby failing to provide a significant temperature gradient between the hot and cold sides. Even when organic TE materials are employed, TEGs still lack sufficient structural freedom. The TE performance of TEGs can be adversely affected when subjected to bending or twisting [159]. In contrast, traditional vertical TEGs, represented by bulk structures, can offer a substantial temperature gradient between the hot and cold sides, primarily due to their construction from rectangular bulk TEG legs. However, the fabrication process for rectangular bulk TEG legs involves complex and time-consuming steps such as powder synthesis, pressing, cutting and polishing, metallization, leg cutting, welding, and module assembly. Additionally, these TEGs lack flexibility, are difficult to integrate, and are challenging to apply on irregular surfaces [156]. Researchers are actively exploring freely formable bulk TE generators. Jungwon Kim et al. [160] prepared porous elastic TE sponges by impregnating polyurethane (PU) sponges with SWCNT ink. It was found that the thermal and electrical properties of the TE sponges are influenced by the void fraction, resistance, and conductive paths, which in turn are affected by pressure and material density. By simple doping, p-type and n-type TE sponges were fabricated and further assembled into a vertically structured sponge TEG (as shown in Figure 10a). The TE power of this sponge TEG increased from 0.17 μW to 2.09 μW under a temperature difference of 55 K, as the compressive strain (ε) increased from 0 to 80%. Traditional BiSbTe-based compounds only achieve high ZT values within narrow temperature gradients. Segmented TEGs, which combine multiple TE elements operating at different temperature ranges, can achieve high efficiency under large thermal gradients. Seong Eun Yang et al. [31] fabricated an all-inorganic high-performance TEG with adjustable peak ZT values across a wide temperature range by sequentially depositing three layers of Bi${\;\;}_{x}$Sb${\;\;}_{2-x}$Te${\;\;}_{3}$ inorganic viscoelastic TE ink using 3D printing and employing Sb${\;\;}_{2}$Te${\;\;}_{42}$ chalcogenide salt as the adhesive at the segments (Figure 10b). As illustrated in Figure 10c, the peak ZT of this TEG extended from room temperature to 250 °C, achieving a record efficiency of 8.7% under a temperature difference of 236 °C. Phillip Won et al. [120] demonstrated a TEG with improved thermal conductivity through an LMEE structure (as shown in Figure 10d).

### 2.3. Other Perspectives on TEG

This section provides a detailed overview of the latest advancements in TE ink technology and the structural design of TEG. It has been demonstrated that elevating the sintering temperature can significantly improve the TE properties of TE inks. However, this enhancement necessitates substrates with higher thermal resilience. Concurrently, augmenting the flexibility of TEGs remains an area of active investigation. Thus, in the concluding section dedicated to the progress in ink-based TE nanogenerators, we will discuss recent innovations in substrate protection, the enhancement of TEG flexibility, and advancements in manufacturing processes.

#### 2.3.1. Substrate Protection

The manufacturing of ink-based TEGs involves printing techniques that are characterized by their simplicity and cost-effectiveness. Following the printing of thick-film ink-based TEGs, the inks typically undergo a sintering process, facilitated by pressure treatment or high-temperature annealing. Post-high-temperature sintering, a decrease in the electrical conductivity of the printed films is often observed, which is usually attributed to the formation of pores within the film or changes in composition due to the sintering of organic materials. Numerous studies have confirmed that increasing the sintering temperature can enhance the electrical conductivity of TEG printed films. However, the high-temperature annealing required for ink sintering poses a challenge for the commonly used flexible polymer substrates, demanding high-temperature resilience [92,123] and necessitating that the process temperatures do not exceed the glass transition temperature of the materials. Nevertheless, the use of substrates with high-temperature stability significantly increases costs in large-scale printing production. Therefore, it is crucial to improve or develop new manufacturing processes that can protect the substrates [161].

Previous studies have established that elevating the sintering temperature and pressure can significantly improve the TE properties of printed TE films [162]. However, the prevalent use of conventional organic substrates poses a challenge, as these materials are susceptible to alterations in their physical properties under elevated processing temperatures. Consequently, research into substrate protection has become a critical need. The primary strategies for substrate protection encompass reducing the processing temperatures, innovating new low-temperature sintering techniques [163,164], and enhancing the thermal resistance of the substrates. The application of mechanical pressure during the thermal curing process enables the production of dense TE films with superior electrical conductivity at reduced processing temperatures. Pin-Shiuan Chang et al. [26] utilized screen printing and pressure sintering methods with Bi-Sb-Te (p-type) and Bi-Se-Te (n-type) inks to craft flexible planar TEGs. In complex manufacturing scenarios, significant challenges arise from high contact resistance, thermal contact resistance, and inadequate thermal coupling due to non-uniform surfaces, which restrict the power output relative to the ZT values of TEG devices and materials [165]. Alloys based on (SbBi)${\;\;}_{2}$(TeSe)${\;\;}_{3}$, whether p-type or n-type, demonstrate remarkable performance at room temperature. Developing p-type and n-type high-performance TE films with comparable synthesis methodologies is crucial for achieving efficient printed TEGs. Md Mofasser Mallick et al. [166] introduced a millisecond photonic curing technique termed “micro-welding”, which safeguards low-temperature flexible substrates, by fusing ball-milled p-BST/n-BT particles with a Cu-Se copper selenide-based inorganic binder, as depicted in Figure 11a. This method not only boosts printability and flexibility but also markedly diminishes the grain boundary resistance among particles, significantly abbreviating the sintering duration. As depicted in Figure 11b, this approach led to the fabrication of flexible, p-type BST (p-BST), and n-type Bi${\;\;}_{2}$Te${\;\;}_{2.7}$Se${\;\;}_{0.3}$ (n-BT)-based printable TE films that exhibit bulk high-performance characteristics. Flexible TEGs tailored for curved surfaces can address the compatibility issues between rigid, brittle TE legs, and heat sources with intricate surfaces. The current body of research on paper-based TEGs with 3D structures is limited, and challenges such as high internal resistance and rigidity remain unresolved. As illustrated in Figure 11c, Xingzhong Zhang et al. [167] demonstrated a high-performance paper-based TEG that can be fabricated without complex preparation involving hot-press sintering, thanks to the optimization of structural design and manufacturing processes.

#### 2.3.2. Flexibility Enhancement

To enhance conformity with heat source surfaces and achieve superior power output, TEGs require significant flexibility to adapt to the contours of irregular heat sources. Furthermore, TEGs with increased flexibility can also mitigate issues associated with material fragility. Departing from conventional single-chain constructs, Dan-Liang Wen et al. [41] developed independent dual chains by screen printing on Bi${\;\;}_{2}$Te${\;\;}_{2.7}$Se${\;\;}_{0.3}$ (n-type) and Sb${\;\;}_{2}$Te${\;\;}_{3}$ (p-type) TE inks, resulting in a dual-chain thermoelectric generator (DC-ThEG) with augmented flexibility, where each chain is capable of harvesting thermal energy (Figure 12a). As shown in Figure 12b, this dual-chain configuration employs silk fibroin materials to cover the inter-chain gaps, acting as a functional layer for detecting liquid water molecules and temperature, thus facilitating multifunctional sensing capabilities. Paper-based TEGs necessitate more efficient and economical manufacturing techniques to enable widespread application. Zuoyuan Dong et al. [168] utilized vacuum-assisted filtration to incorporate doped Bi${\;\;}_{2}$Te${\;\;}_{3}$ and Sb${\;\;}_{2}$Te${\;\;}_{3}$ into modified cellulose paper, creating a conductive path through the particle deposition layer on the paper surface and the particle penetration layer within the paper. Demonstrated in Figure 12c, a foldable paper-based TEG was fabricated using copper foil to connect the modified paper sections, alongside a fingertip touch sensor, validating the feasibility of employing modified cellulose paper for fabricating paper-based TEGs and thermal sensors. Christos K. Mytafides et al. [138] successfully assembled a flexible, metal-free, all-carbon organic TEG by tandemly connecting p-type and n-type SWCNT TE elements that exhibit outstanding electrical conductivity (Figure 12d).

#### 2.3.3. Novel Ink-Based TEG Fabrication Processes

Optical 3D printing technology, recognized for its mask-less AM capability, enables the cost-effective creation of high-resolution patterned architectures. However, the choice of materials for photo-curable resins limits the functionality and application scope of the printed objects. ChaMs inks emerge as ideal candidates for fabricating patterned structures. As illustrated in Figure 12e, Seongheon Baek et al. [169] introduced a universal optical printing technique based on Digital Light Processing, utilizing a photoacid generator to develop a photo-curable ChaM-based inorganic ink. Employing a two-step patterning approach, this method facilitated the fabrication of patterned microscale TEGs. This technique broadens the application of inorganic semiconductor materials as optically printable substances, demonstrating the substantial potential of directly constructing 2D and 3D structures from inorganic materials. A significant challenge lies in synthesizing inorganic nanoparticle inks from exfoliated 2D materials and achieving solid formations post-sintering that exhibit performance akin to that of single-crystal 2D materials. Stéphane Jacob et al. [170] showcased a novel fabrication process for mixed polycrystalline TiS${\;\;}_{2}$/hexylamine films, enhancing the process through electrochemical intercalation. By electrochemically embedding hexylammonium (HA${\;\;}^{+}$) cations and dimethyl sulfoxide (DMSO) solvent into TiS${\;\;}_{2}$ single crystals, an n-type hybrid TiS${\;\;}_{2}$/organic superlattice material was synthesized, followed by solvent exchange with H${\;\;}_{2}$O to obtain TiS${\;\;}_{2}$[(HA)${\;\;}_{x}$(H${\;\;}_{2}$O)${\;\;}_{y}$(DMSO)${\;\;}_{z}$]. Subsequently, high-boiling-point, high-molecular-weight tetra-n-butylammonium was used to replace the solvent intercalated with HA${\;\;}^{+}$, improving the power factor and reducing the thermal conductivity. The polycrystalline TiS${\;\;}_{2}$ powder prepared via solution processing, followed by mechanical intercalation of HA (Figure 12f), resulted in an increase in power factor for the 2D TE inorganic nanomaterial film, surpassing the TE performance of organically intercalated TiS${\;\;}_{2}$ single crystals. The TE power factor of printed TiS${\;\;}_{2}$/hexylamine films far exceeded that of single crystals, and the printed TEG based on eight TiS${\;\;}_{2}$ legs exhibited a high power factor, laying the foundation for Low-Power Wide-Area Network development anticipated in the IoT era. 

## 3. Ink-Based TENGs

TENGs have demonstrated significant potential in the collection of low-frequency and weak mechanical energy, providing power for distributed wireless sensors [12,171]. The advent of printing technology has enabled the large-scale manufacturing of ink-based nanogenerators to become feasible. By integrating multi-step 3D printing processes that combine 3D printing technology with extrusion deposition techniques, it is possible to achieve programmable layer-by-layer manufacturing of TENGs [172]. Recently, Hui Li et al. [173] extensively discussed the latest advancements in printed TENGs from two primary perspectives: ink preparation and printing methodologies. They also highlighted that current challenges facing ink-based TENGs, such as the printability of triboelectric inks, the complex structural design of TENGs, and printing resolution, still need to be addressed. In this section on “Ink-Based TENGs”, the focus will be on the materials for TENG inks, the latest designs of ink-based TENG structures, and textile-based printed TENGs, reviewing and summarizing the latest progress in the field of ink-based TENGs.

### 3.1. Optimization of TENG Ink Materials

In the domain of ink-based TENGs, the choice of triboelectric ink materials plays a pivotal role in determining the electrical output performance of the generators [174]. Therefore, this section is dedicated to discussing the recent developments in triboelectric ink materials. The discussion initiates with conductive inks employed for electrodes, specifically focusing on carbon-based composite materials for their conductive properties. Following this, recent progress in both positive and negative triboelectric inks will be examined. The discourse concludes by addressing the application of specialized functional inks that have been successfully incorporated into ink-based TENGs. Detailed performance metrics of these inks can be found in Table 1.

#### 3.1.1. Electrode Conductive Ink

Research in the domain of conductive inks is crucial for advancements in energy harvesting and storage, with the development of functional nanoin-ks being predicated on the foundation of conductive inks through the integration of nanofillers. DIW, recognized for its efficiency and cost-effectiveness, stands out as a prominent extrusion-based printing technique. Ting Huang et al. [183] have concentrated their efforts on the exploration of conductive materials fabricated via DIW, providing a comprehensive review of the rheological properties and formulations of graphite-based, Mxene-based, and CNT-based printable inks, which are among the most extensively utilized electrode materials. Their discussion extends to the application of these materials in rechargeable batteries and supercapacitors while also identifying the existing challenges related to the printability, functionality, and resolution of conductive inks. In a parallel vein, Y.Z.N. Htwe et al. [184] delved into graphene-based inks, examining the forefront of research on graphene and its composite conductive inks. This investigation also encompasses various flexible, stretchable polymers, and paper-based and textile-based substrates, elucidating their distinct properties and benefits. Furthermore, it presents an analysis of the obstacles and prospective directions for the development of eco-friendly, high-performance conductive inks, offering valuable insights into the future trajectory of this field.

Composite materials are capable of endowing functionalities and fluid properties that single materials alone cannot achieve, where the role of conductive nanofillers is critical in dictating the electrical and mechanical performance of nanocomposites. The integration of graphene with NPs and NWs, particularly in graphene nanoparticle composites, has been shown to exhibit superior electrical conductivity among other advantageous properties. Present research efforts are concentrated on enhancing the electrical characteristics of carbon-based inks, improving their performance under specific conditions, and addressing the challenges associated with ink aggregation. For example, Guanjun Zhu et al. [185] fabricated multifunctional films through the screen printing of carbon ink mixed with silver nanowires/thermoplastic polyurethane (C-AgNW/TPU) and integrated them with polytetrafluoroethylene (PTFE) membranes to develop a single-electrode mode TENG for energy collection. Devices that are self-powered and operate in harsh environments require electrode materials that possess inherent waterproofing and chemical inertness [186]. In this context, Guisong Yang et al. [176] developed a corrosion-resistant fluoropolymer/carbon nanotube (FP/CNT) material that is both chemically inert and superhydrophobic (Figure 13a). They then used DIW to construct mechanically robust FP/CNT electrodes, subsequently assembling a droplet-based electricity generator that can illuminate 50 LEDs with a single droplet, as depicted in Figure 13b. The dispersible nature of Laponite’s discotic crystals facilitates the formation of interconnected networks with polymer chains. Poly(ethylene oxide) (PEO), known for its excellent water solubility, can absorb NPs and physically interlink with Laponite to form a gel. Drawing inspiration from sesame candy, as illustrated in Figure 13c, Zhuang Li et al. [32] employed Laponite to modify CNTs and merge them with PEO, along with PEDOT:PSS to forge a conductive network, yielding a viscoelastic putty-like conductive nanocomposite (LPPC). The self-healing viscoelasticity of LPPC can be modulated by adjusting its water content. Represented by Ti${\;\;}_{3}$C${\;\;}_{2}$T${\;\;}_{x}$, MXene materials are recognized for their hydrophilicity, conductivity [187], and flexibility, with their abundant functional groups ensuring satisfactory dispersion stability in solvents. However, the printing of MXene electrodes is impacted by self-stacking effects, attributed to the van der Waals forces between MXene layers. Dong Wen et al. [177] mitigated this self-stacking by intercalating 1 wt% graphene nanosheets between the layers of printed MXene (Figure 13d), thus expanding the interlayer distance and increasing the specific surface area of MXene. Tin (Sn), prized for its economic value and low melting point, is prone to rapid oxidation during annealing, and graphene modified by Sn is more susceptible to aggregation. Omar Kassem [178] synthesized uniformly sized Sn NPs via an enhanced polyol process and utilized intense xenon light technology to induce local irradiation of inkjet-printed Sn NPs, causing them to contract and form local carbon precursors. This process transformed the printed Sn NPs into Sn@graphene (Sn@G) core–shell structures, ultimately leading to the fabrication of high-performance Sn@G material electrodes.

Currently, the conductivity of p-type polymer inks significantly surpasses that of n-type polymers [188,189]. Furthermore, during the sintering process, the differential evaporation rates between the edges and the center of ink droplets give rise to the coffee ring effect, potentially leading to discontinuous conductive pathways and severely compromising conductivity [190]. Chi-Yuan Yang et al. [33] has reported an alcohol-based n-type BBL:PEI polymer conductive ink with high conductivity, observing that at a PEI concentration of 50 wt%, conductivity reaches a saturation point at 7.7 ± 0.5 S cm${\;\;}^{-1}$. A brief treatment at 150 °C for 5 min is sufficient to achieve an n-type conductivity of 1 S cm${\;\;}^{-1}$, whereas annealing for 2 h results in a peak conductivity of 8 S cm${\;\;}^{-1}$, providing a solution to the pressing issue of achieving complementary hole and electron transport. Wenxiu Wu et al. [179] has developed high-conductivity inks featuring 1.65 μm short Ag NWs with a reduced sintering temperature, by adjusting the reaction rate and lowering the chloride ion concentration using the growth regulator NaCl. Additionally, Ag NWs serve to attenuate the coffee ring effect.

#### 3.1.2. Friction Dielectric Layer Ink

In the development of ink-based TENGs, the use of conductive inks for electrode fabrication is complemented by the employment of positive and negative triboelectric inks for the creation of functional layers. A significant area of research is focused on the enhancement of the positive triboelectric characteristics of positive triboelectric inks [174]. All-inorganic metal halide perovskites (CsPbX${\;\;}_{3}$, X = Cl, Br, I) are recognized for their cost-effectiveness and photophysical attributes [186,191,192], albeit with limitations in terms of solid yield and stability. As depicted in Figure 14a, Long Chen et al. [193] have successfully synthesized metal halide solids CsPbBr${\;\;}_{3}$@KBr characterized by a narrow full-width at half-maximum, elevated photoluminescence quantum yield, and enhanced stability through a direct water-evaporation crystallization technique. These solids act as fillers for the positive-friction-layer polyvinyl alcohol (PVA) films in TENGs, facilitating the development of colorless, transparent, aqueous positive triboelectric inks. Utilizing the CsPbBr${\;\;}_{3}$@KBr/PVA layer, a self-powered wearable photoluminescent sensor was engineered. Polylactic acid (PLA), known for its eco-friendliness yet limited by mechanical properties and water solubility, benefits from the addition of high-performance additives. Agneyarka Mohapatra et al. [194] enhanced the positive triboelectric functionality of the PLA friction layer by incorporating carbon nanotube–zinc oxide (CNT-ZnO) hybrid core–shell nanostructures as additives. Together with an acrylonitrile butadiene styrene (ABS) friction-negative layer and leveraging Fused Deposition Modeling (FDM)-based 3D Printing (3DP) technology, a sophisticated ring-patterned TENG was fabricated, with investigations into the impact of layer thickness and fill rate on the performance of the TENG. Moreover, Guoxu Liu et al. [195] achieved the optimization of the positive triboelectric performance by doping the Dragon Slow Skin 10 (DSS10) substrate with nickel powder in varying proportions, culminating in the formulation of functional printed inks. The phenomenon of charge screening can lead to a reduction in the frictional potential between materials, subsequently lowering the energy conversion efficiency of TENGs. Presently, there is a noticeable dearth of research aimed at alleviating charge screening and minimizing the loss of frictional charges by introducing interface layers. SWCNTs stand out for their exceptional mechanical strength, stability, and electrical conductivity. In this context, Byeong-Cheol Kang et al. [196] have successfully mitigated frictional charge losses and significantly enhanced the output performance of PDMS-based TENGs by forming a positively charged layer of a random SWCNT network on the surface of Al electrodes using rod printing technology. Importantly, the spatial distribution and density of the discrete SWCNT random networks can be precisely controlled by varying the number of printing layers.

Current research on negative triboelectric inks primarily focuses on the performance enhancement of polyvinylidene fluoride (PVDF) materials. As a non-toxic material, PVDF possesses high negative triboelectricity, flexibility, wear resistance, and corrosion resistance. Notably, ferroelectric PE PVDF materials with high crystallinity can significantly improve the performance of TENGs [197]. PVDF typically exists in the α-phase, but the β-phase of PVDF can significantly enhance the electrical output of TENGs, necessitating the induction of a phase transition from α-phase to β-phase [198]. Bhavna Hedau et al. [34] have improved the crystalline structure of PVDF by incorporating MoS${\;\;}_{2}$ through solution processing, enhancing charge capture capabilities through the synergistic effect of MoS${\;\;}_{2}$’s triboelectric trap states and promoting the phase transition of PVDF from α-phase to the more polar β-phase. Additionally, they fabricated a self-polarized MoS${\;\;}_{2}$@PVDF hybrid nanocomposite film with enhanced triboelectric effect using a rod-coating printing technique, as depicted in Figure 14b, thereby constructing TENGs with superior performance. Muhammad Tayyab et al. [199] employed electrospinning technology with different growth durations to cultivate high-conductivity PVDF NFs based on NFs over time and introduced printer ink (PI) nanofillers to improve crystallinity, as shown in the microscopic structure of the ink in Figure 14c. This improvement increased the output of PVDF-PI NFs-based TENGs, further enhancing the performance due to the parallel alignment of ferroelectric domains.

**Figure 14 ijms-25-06152-f014:**
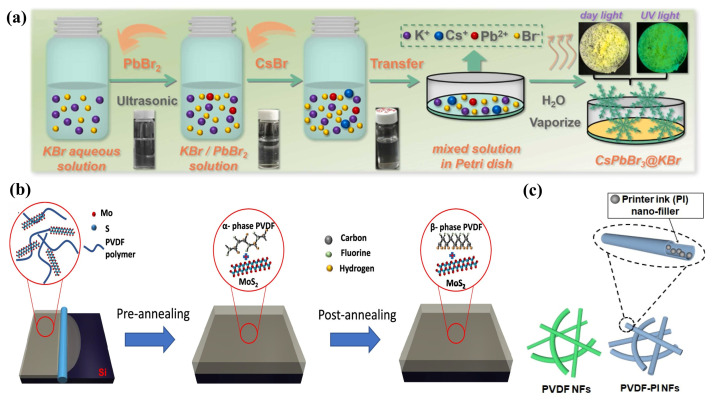
Preparation of triboelectric ink and its microstructure. (**a**) Preparation of CsPbBr${\;\;}_{3}$@KBr metal halide solids and their transparent aqueous ink. Reprinted with permission from [193]. Copyright 2023, Elsevier. (**b**) Fabrication of the MoS${\;\;}_{2}$@PVD hybrid nanocomposite film and its ink crystal structure. Reprinted with permission from [34]. Copyright 2022, American Chemical Society. (**c**) Schematic of the microstructure of PVDF-PI NFs. Reprinted with permission from [199]. Copyright 2020, Elsevier.

#### 3.1.3. Special Function Inks

By chemically modifying the functional groups of molecular materials based on solutions, the surfaces of triboelectric layers can be designed at low cost and with simplicity, thereby enhancing the triboelectric charge density [200]. Currently, inks with special functionalities primarily focus on performance optimization in terms of ink elasticity and the promotion of triboelectric charge transfer. Silicone rubber, known for its mild vulcanization conditions and excellent mechanical properties and biocompatibility, is widely used [201]. High-viscosity, shape-retaining silicone rubber liquid precursors, prepared through vulcanization DIW, enable the 3D printing of silicone elastomer inks with specific structures required for reciprocating operations. As shown in Figure 15a, Renhao Zheng et al. [202] utilized PDMS micro-powder with high electron affinity as a thixotropic agent, mixing it with various ratios of PTFE silicone rubber elastomer precursors to develop PTFE/PDMS silicone elastomer composite inks with adjustable mechanical strengths, as depicted in Figure 15c. This led to the construction of high-performance TENGs equipped with PTFE/PDMS elastic support structures. Gallium-based LM polymers or LMEE composites, composed of a polymer base and LM alloy, offer stretchability, conductivity, and mechanical properties [203]. Phillip Won et al. [120] prepared LMEE inks based on an EGAIN-PDMS emulsion. Additionally, gel transfer shockwaves were used to render the electrically insulating LMEE structures, written via DIW, conductive. Finally, as illustrated in Figure 15b, TENGs enhanced with LMEE structures for improved charge transfer were demonstrated. Surface modification can be achieved through molecular electrostatic self-assembly strategies [204]. Jia-Ruei Yang et al. [35] modified the surface of PDMS dielectric layers (Figure 15d) with electrostatically self-assembled molecules of 1H,1H-perfluorooctylamine (F${\;\;}_{15}$-NH${\;\;}_{2}$), combined with PDINI-Modified Ag electrode layers (Figure 15e), to fabricate stable, high-performance TENGs. During this process, the PDMS surface adsorbed F${\;\;}_{15}$-NH${\;\;}_{2}$ through electrostatic interactions with the protonated amine groups of F${\;\;}_{15}$-NH${\;\;}_{2}$. Concurrently, during film formation, the accumulation of highly electronegative perfluoroalkyl chains at the air interface induced surface dipoles, thereby promoting electron transfer between the electrode and dielectric layer.

### 3.2. TENG Structure Design

Beyond the choice of TENG ink materials, the structural design of TENGs is another critical factor affecting their electrical performance [171,205,206]. While previous reviews have predominantly revisited the latest progress in ink-based TENGs through the lens of printing technologies, they have often overlooked an in-depth summary and exploration of the structural design of ink-based TENGs. This section will address the structural design of TENGs, beginning with planar ink-based TENG structures that include micro-surface and linear surface structures. Subsequently, it will delve into 3D ink-based TENG structures, primarily focusing on protruding and block structures. Lastly, the discussion will extend to some specialized TENG structures designed for pulse output.

#### 3.2.1. Planar Ink-Based TENG Structure

A straightforward printing method for fabricating flexible TENGs is necessitated by the demand for ease in manufacturing. Creating charge traps at the interface between the friction layer and electrodes offers a simple and cost-effective approach to enhance the surface charge density of the friction layer, thereby improving TENG performance [207,208]. Fluoroelastomer (FKM) polymers are known for their electronegativity, stretchability, and durability [209]. However, research on surface enhancement of FKM and its straightforward integration with highly conductive electrodes is lacking. Injamamul Arief et al. [180] have addressed this by manipulating the porosity and depth of FKM to form periodic porous structures on its surface. By curing the FKM substrate of the negative friction layer during transfer printing and employing a TiO${\;\;}_{2}$ deposition-based embossing technique, they fabricated a highly porous micropatterned surface, as illustrated in Figure 16a. Concurrently, CO${\;\;}_{2}$ laser-induced graphene (LIG) electrodes were transfer-printed to construct high-power single-electrode TENGs. Hui Li and colleagues [181] achieved wrinkled silver electrodes by inkjet printing silver electrodes while pre-stretching the substrate. Furthermore, a contact-separation mode TENG was developed by combining directly written silicone layers. Additionally, external silicone encapsulation protected the silver electrodes (as shown in Figure 16b) and provided tribo-negative electrons capable of instantaneously lighting up 20 LEDs.

Beyond the exploration of surface voids and wrinkles, linear surface microstructures on planar ink-based TENGs have garnered significant attention. E Cheng et al. [37] devised a method to create negatively charged frictional Ag NWs (HPMC) ink by blending Ag NWs with hydroxypropyl methylcellulose (HPMC) in an aqueous solution. Following this, AgNW linear arrays were meticulously printed on the positively charged friction layer of PDMS utilizing electrohydrodynamic jet printing technology. As illustrated in Figure 16c, these arrays made contact with another PDMS layer featuring microgrooves, thereby enhancing internal friction. Furthermore, the strategically designed charge trap structure on the PDMS/Ag NWs/PDMS surface served to amplify external friction. This orchestrated synergy between internal and external friction mechanisms significantly augmented TENG performance, culminating in a tenfold enhancement in output power relative to conventional flat TENGs. This advancement enabled the fabrication of a MATENG capable of charging a 22 μF capacitor to 3 V within 117 s. In the realm of durability enhancement, self-healing and shape memory polymer-based self-recoverable TENGs have proven to be beneficial. As depicted in Figure 16e, objects created through 4D printing technology—which integrates 3D printing with shape memory materials—demonstrate dynamic evolution in response to external stimuli. In this vein, Long-Biao Huang et al. [182] introduced a transparent self-recoverable 4D-printed TENG by synergizing 4D spray printing of shape memory polymers (SMP) with printed AgNW electrodes, as showcased in Figure 16d. Gallium-based LM polymers or LMEE composites, which combine a polymer base with a LM alloy, are noted for their stretchability, conductivity, and mechanical robustness. Phillip Won et al. [120] highlighted a TENG enhanced by LMEE structures facilitating charge transfer, employing EGAIN-PDMS emulsion LMEE ink, as demonstrated in Figure 16f.

#### 3.2.2. Three-Dimensional Ink-Based TENG Structure

The integration of nanostructures on the surfaces of active materials has been demonstrated to significantly enhance the surface contact area, which in turn facilitates increased charge transfer and efficiency [200]. In the realm of 3D ink-based TENGs, research has predominantly concentrated on exploring structures such as hemispherical, pyramid-like, and block forms. These 3D surface structures are instrumental in elevating the surface charge density, with AM technologies offering a promising pathway for the scalable production of intricate 3D constructs. For example, Hui Li et al. [36] have notably advanced the performance of micro-top-layer structure TENGs by ingeniously fabricating jet-printed micro-droplet top-layer structures on the friction layer. This was achieved by meticulously adjusting ink properties, printing process parameters, and substrate wettability to create tunable microstructure arrays, as depicted in Figure 17a. Despite these advancements, the field still faces a paucity of precise investigations that conclusively demonstrate the impact of increased surface area on TENG performance. Ken C. Pradel et al. [210] addressed this by fabricating interlocking hemispherical arrays of electropositive polyimide and electronegative PVDF materials through thermal micro-contact printing. By modulating the spacing of pattern features to adjust surface contact levels, they achieved a performance enhancement several times greater than that of flat samples, thereby deepening the understanding of TENG surface interactions, as showcased in Figure 17b. Further contributing to the field, Mingzhu Zhu et al. [211] introduced an enhanced soft finger incorporating a single-electrode triboelectric curvature sensor. This was accomplished by directly 3D printing the active layer of S-TECS onto the finger body’s surface and straightforwardly assembling it with a stretchable electrode coupled with PDMS. Remarkably, this configuration attained an exceptional soft finger-bending curvature sensitivity of up to 8.2 m${\;\;}^{-1}$ at an ultra-low working frequency of 0.06 Hz, as illustrated in Figure 17c. The 2D MXene (Ti${\;\;}_{3}$C${\;\;}_{2}$T${\;\;}_{x}$) materials, known for their exceptional conductivity, high charge carrier mobility, robust mechanical properties, stability, and highly electronegative surface [212,213], were utilized alongside Ecoflex (a silicone rubber), another highly electronegative material, by Shipeng Zhang et al. [39]. This collaboration led to the construction of pyramid arrays on MXene/Ecoflex nanocomposite materials through 3D printing technology, culminating in the design of a self-powered, single-electrode, simple-structured ring-shaped triboelectric sensor, as shown in Figure 17d. Addressing the challenge of stably harvesting micro-flow energy in deep ocean currents with speeds less than 10 cm s${\;\;}^{-1}$, a domain where traditional electromagnetic generators and PENGs are less effective, Yan Wang et al. [38] developed a simple underwater flag-like triboelectric nanogenerator with superior low-speed performance. This innovative design employs a two-layer conductive ink film of polyethylene terephthalate (PET) and polytetrafluoroethylene, with the ink layer’s edges sealed by waterproof PTFE tape to prevent water contact with the triboelectric layer. Additionally, a cylindrical structure was utilized to induce a Kármán vortex street, thereby enhancing the UF-TENG’s vibration, as evidenced in Figure 17e.

Non-contact sensors, specifically Proximity Non-Contact TENGs, are emerging as a new trend in the field of human–machine interaction due to their ability to avoid mechanical fatigue and sense over longer distances. These sensors generate charges through electrostatic induction without the need for direct contact, despite challenges related to the loss of triboelectric electrons. To address this issue, Binbin Wang et al. [40] utilized freeze-drying assisted 3D printing technology to fabricate a CNF/MXene Proximity Non-Contact TENG capable of multidirectional non-contact motion detection. As shown in Figure 17f, the device’s deep-well hierarchical structure is capable of preserving induced charges over an extended period. Moreover, the hydrogen bonding between MXene (Ti${\;\;}_{3}$C${\;\;}_{2}$T${\;\;}_{x}$) nanosheets and CNF not only enhances the dispersibility of the viscoelastic ink but also its high charge capture ability helps to prevent charge loss. This high charge capture capability also allows the printed CNF scaffold to have more open pores, further increasing charge generation.

#### 3.2.3. Pulse Output Ink-Based TENG Structure

Unlike traditional TENGs, pulsed TENGs, characterized by low internal equivalent resistance, not only achieve high output but are also easily impedance matched with Power Management Circuits. The introduction of Synchronous Trigger Mechanical Switches (STMS) in series allows for the activation of pulsed switching. There is a lack of research on STMSs designed for rotating antennas as rotating TENGs. Wanyu Shang et al. [214] integrated STMSs onto the triboelectric layer using printing technology, successfully fabricating a rotational freestanding triboelectric-layer mode pulsed-TENG. This generator is capable of producing alternating current and UD Radio Frequency pulses based on STMSs and can easily match with rotating frequencies to achieve high output. Furthermore, the system incorporates a gearbox and a passive Power Management Circuit to form a self-powered system, demonstrating significant application potential.

### 3.3. Other Perspectives on TENG

#### 3.3.1. Integration with NFs

Fiber-based TENGs are distinguished by their lightweight, flexibility, and durability [215]. Moreover, the extensive contact area offered by nanofiber interfaces significantly enhances TENG output. Nevertheless, the electrical performance of textile-based TENGs under tensile conditions necessitates further enhancement. Despite the limited stability of Mxene materials, their abundant functional groups position them as ideal candidates for balancing the conductivity and stretchability of conductive fibers. The development of high-performance TENGs compatible with textile technologies presents ongoing challenges. Yi Hao et al. [216] have achieved antioxidative P-MXene ink by modifying the surface of Mxene with in situ polymerized dopamine. Following this, the ink was co-polymerized with highly stretchable MXene/TPU (MP) fibers in situ, yielding MMP composite fibers characterized by high conductivity (4.32 S cm${\;\;}^{-1}$), an extensive deformation range (675% strain), and notable mechanical strength (3.76 MPa). Utilized as the negative triboelectric layer in TENGs, these fibers demonstrated a peak V${\;\;}_{OC}$ of 21.07 V, a I${\;\;}_{SC}$ of 0.92 μA, and an output power density of 0.16 mW m${\;\;}^{-2}$ at 3 Hz. Jaeho Kim et al. [217] prepared the positive triboelectric layer of nylon-6 TENGs by spin-coating nylon-6 onto conductive Ni-Cu fabric, with the negative layer composed of barium titanate NPs (BaTiO${\;\;}_{3}$ NPs) and PDMS surface-modified 1D poly[styrene-isoprene-b-styrene] tubes, leading to the construction of high-stretch TENGs. Satyaranjan Bairagi et al. [218] created a positive triboelectric layer by electrospinning nylon-66 NFs onto silk fabric surfaces and constructed the negative triboelectric layer by coating PET fibers with PVDF ink, producing a high-performance t-TENG with a peak power density of 280 mW m${\;\;}^{-2}$. The positive triboelectric properties of CNFs were enhanced by grafting PEI onto CNFs. Pei-Yong Feng et al. [219] introduced CNTs and branched PEI onto the fabric surface through solution processing combined with a polyamidation reaction, forming a negative triboelectric layer. This TENG achieved a power density of 3.2 W m${\;\;}^{-2}$ at an external resistance of 5 × 10^6^
Ω. Ismael Domingos et al. [220] employed three printing deposition methods—droplet, immersion, and spray (as shown in Figure 18a)—alongside textile flattening techniques to fabricate soft, shape-adaptable graphene droplet films, graphene immersion films, and graphene spray films electrodes. By depositing a PDMS layer and utilizing the polyester/polyamide fabric as the active material, textile-embedded TENG sensors were manufactured.

TENGs predominantly generate pulsed alternating current, necessitating rectification to derive direct current capable of directly powering electronic devices. Consequently, the development of DC-TENGs is a key area of current research focus. Furthermore, the prospect of simplifying TENG structures paves the way for their multifunctional integration. Yiding Song et al. [222] have delved into the working mechanisms of various DC-TENGs, emphasizing the analysis of crucial factors influencing the output performance of DC-TENGs and outlining potential directions for future advancements. Despite the high internal resistance characteristic of conventional TENGs, metal/inorganic semiconductor friction can produce high DC power output with low internal resistance on the surfaces of inorganic semiconductors. However, research on durable, high-output, and flexible DC TENGs, particularly those employing pure organic semiconductor materials such as PEDOT:PSS, polypyrrole, or PANI, remains scarce. As illustrated in Figure 18b, Guoxu Liu et al. [221] devised a DC flexible textile organic TENG by integrating PEDOT:PSS and PVA blend films, hydrophilic conductive inks, hydrophobic conductive textiles, and an Al slider. In the presence of frictional energy, the substantial quantity of electron–hole pairs generated at the metal–semiconductor interface within the OTG will undergo directional flow under the electric field, thereby generating a current (as shown in Figure 18c). Additionally, the incorporation of PVA markedly improves both the electrical and mechanical properties. Hydrogels, with their inherent ionic conductivity, have attracted extensive research interest. Yan-Yuan Ba et al. [223] have derived a CNF film with hydrogel characteristics from cellulose extracted from carrots. By employing asymmetric pairing effects and enhancing the CNF film (ion-CNF) with CaCl${\;\;}_{2}$ ion doping, they have conceptualized two types of TENGs using ion-CNF for both the positive triboelectric layer and as a combined negative triboelectric layer cum electrode, unveiling a novel hybrid triboelectric working mechanism that integrates surface charge transfer with ion migration.

#### 3.3.2. Novel Ink-Based TENG Fabrication Processes

Traditional manufacturing methods encounter difficulties in fabricating highly flexible TENGs with intricate geometric configurations. Concurrently, composite materials based on PDMS are preferred for their chemical stability, flexibility, and biocompatibility. PDMS featuring hierarchical microstructures demonstrates superior hydrophobic properties and low surface energy [224]. Qi-Jun Sun et al. [225] employed rod-assisted printing to produce graphite/PDMS composite triboelectric films with hierarchical microstructures, enhancing the electrical output of TENGs by fine-tuning the graphite filler concentration within PDMS, thus achieving high resilience to various liquids. Despite nano-fibers’ suitability for TENGs due to their flexibility and extensive surface area, advancements towards their deformability and enhanced durability remain essential. As depicted in Figure 19a, Yi Li et al. [226] synthesized physically interlocked PVDF-HFP/SEBS (PHS) fiber films that exhibit hydrophobicity and elasticity by concurrently electrospinning poly(vinylidene fluoride-co-hexafluoropropylene) (PVDF-HFP) and electro-spraying styrene-ethylene-butylene-styrene (SEBS). The electrodes, composed of liquid metal (gallium indium tin particles)/silver flakes (LM/Ag), were fabricated through screen printing (shown in Figure 19b), culminating in the assembly of a stretchable, breathable, and hydrophobic nano-fiber-based TENG, as illustrated in Figure 19c. This highlights the significant potential of integrating electrospinning and electro-spraying technologies for producing high-performance nano-fiber films. Furthermore, screen printing technology enables the customization of patterns for textile-based TENGs. Chi Zhang et al. [227] created a durable, screen-printed textile TENG by sequentially printing nylon and silver inks onto fabric (as shown in Figure 19d), with the nylon ink base layer exhibiting strong adhesion to the fabric. Additionally, fabricating flexible single-electrode TENGs with multifaceted structures from diverse materials in a single-step process poses a challenge. As shown in Figure 19e, Zhenwei Wang et al. [228] developed a high-performance, shape-adaptable, fully flexible single-electrode TENG using a one-pot coaxial DIW technique, simultaneously printing viscoelastic silicone elastomer (PDMS) ink to form the outer shell of the triboelectric layer and a silicone/carbon black flexible electrode core (as shown in Figure 19f).

## 4. Other Ink-Based Nanogenerator Technologies

In the preceding sections of this manuscript, an in-depth exploration has been undertaken regarding two primary classifications of ink-based nanogenerators—namely TEGs and TENGs—emphasizing the forefront of developments in ink materials and structural configurations. Expanding upon this foundational discourse, this section aims to broaden the scope to encompass three additional variants of ink-based nanogenerators: PE, moisture-enabled, and hybrid nanogenerators. It will meticulously examine the cutting-edge advancements in the realms of material selection and structural design pertinent to these technologies [229,230,231]. This section is dedicated to systematically synthesizing the evolutionary trajectories of these three categories of ink-based nanogenerators, with the objective of furnishing a more holistic and in-depth perspective on the ongoing research within this specialized field.

### 4.1. Ink-Based PENGs

PENGs exhibit significant advantages over TENGs due to their immunity to temperature fluctuations, stray capacitances, and sliding mechanisms. Notably, PVDF, characterized by its hydrophobic surface, stands out for its superior PE, pyroelectric, and ferroelectric attributes. The electroactive multiphasic nature of PVDF is ascribed to the specific orientation of its macromolecular chains. Advanced microstructural design can further enhance the efficacy of PVDF-based PE devices. PVDF-HFP, as a PE polymer material, embodies flexibility, transparency, and robust mechanical properties, rendering it an exemplary choice for fabricating high-performance PE devices due to its intrinsic permanent electric dipole moment in the polar β-phase. However, devising a straightforward methodology to produce PVDF-HFP with a high content of the electroactive β-phase remains a challenge. In a notable study, Hai Li et al. [42] achieved this by introducing the hydrophobic agent 1H, 1H, 2H, 2H-perfluorodecyltriethoxysilane (PFOES), into the PVDF-HFP matrix to facilitate hydrogen bond interactions, coupled with the use of perfluorooctanesulfonic acid as a nucleating agent to induce strong hydrogen bonding, thereby effectively promoting the β-phase formation (Figure 20a). This approach led to the creation of PVDF-HFP ink with a high β-phase content. A high-performance, self-polarized, transparent ST-PENG was subsequently developed, employing indium tin oxide—polyethylene terephthalate electrode layers and PVDF-HFP active PE layers. Additionally, the advancement in unsupported structure PE ceramics significantly boosts the electromechanical responsiveness of PE devices. Despite the complexity and high precision requirements, Zhouyao Li et al. [44] successfully developed a DIW compatible suspension using organic PZT powder (Figure 20b). Leveraging the DIW technique, along with the malleability of the green body to undergo shape modification and a secondary shaping process, this innovation enabled the fabrication of complex unsupported structure PE ceramics with commendable PE properties, thereby overcoming the inherent limitations of DIW and broadening the avenues for manufacturing geometrically complex PE ceramics with macroscopic structures. Despite a paucity of research on the PE-photovoltaic electronic effects in water-processable all-organic materials, Hari Krishna Mishra et al. [43] made a significant breakthrough by procuring electroactive δ-phase PVDF NPs within a ternary phase system of PVDF-DMF-water through a straightforward solution-processing phase-separation technique (Figure 20c). By further surface-modifying the δ-PVDF NPs with in situ copolymerized polydopamine-polyethyleneimine (PDA-PEI), a water-dispersible ink was obtained, exhibiting exceptional ferroelectric and PE properties and demonstrating PE-photovoltaic effects under light irradiation and mechanical stress. This innovation not only facilitated the development of self-powered photodetectors and acoustic sensors with superior voice-signal-recognition capabilities but also opened a novel research pathway for the application of PVDF NPs in PE ink formulations.

PVDF is widely recognized for its exceptional PE properties, making it a pivotal material in the construction of standard vibration energy harvesters. The performance of these harvesters can be significantly enhanced by employing multilayer film configurations to increase the power density of energy collection. Nicolas Godard et al. [45] achieved a milestone by screen-printing ten layers of poly(vinylidene fluoride-co-trifluoroethylene) (P(VDF-TrFE)) films on a poly(ethylene naphthalate) (PEN) polymer substrate, interspersed with PEDOT:PSS interlayers, culminating in a fully printed multilayer P(VDF-TrFE) PE polymer with performance comparable to that of the finest PE ceramics (Figure 20d). However, PVDF treated with low-vapor-pressure solvents presents challenges in compatibility with fabrication processes such as FDM, Stereolithography, and Solvent Cast 3D Printing. As depicted in Figure 20e, Hai Li et al. [232] innovated by employing Solvent-Assisted Direct-Write 3D Printing, exposing high β-phase content self-polarizing PVDF ink to a nonsolvent environment (ethanol, for instance) during printing. This method facilitated effective ink concentration adjustment and rapid precipitation of porous structures, with further investigations revealing that cylindrical fiber-based PENGs exhibit superior PE performance compared to their elliptical counterparts. Balancing the rheological properties and printability of PE inks poses a significant challenge. Sk S. Hossain et al. [233] identified the optimal ink formulation by evaluating different solid volumes of PZT ink within an aqueous binder system composed of PVA, glycerol, and PZT powder, providing effective solutions to defects encountered in various 3D printing processes (Figure 20f,g). Moreover, the application of Cold Isostatic Pressing to enhance the green body density led to the fabrication of high-density 3D printed PZT ceramics. In 3D printing technologies, the incorporation of multi-tier porous structures can enhance the flexibility of structures with high PE ceramic content. Nonetheless, a straightforward method for fabricating 3D porous structures that simultaneously optimize polarization efficiency and electrical amplification effects remains elusive. Inspired by thermally expandable play bricks, Fang Chen et al. [46] reported the fabrication of a PENG capable of generating up to 150 V of V${\;\;}_{OC}$ and 16 μA of I${\;\;}_{SC}$ under jumping motions, by alternately layering printed polydimethylsiloxane/barium titanate PE ink layers with radially distributed PDMS/CNTs conductive ink layers within the layers (Figure 20h). In this structure, the high content of conductive filler BT significantly enhanced the polarization electric field. Furthermore, the introduction of voxelated isolated conductive networks through Multi-material 3D Printing greatly optimized the polarization pathways and homogenized the electric field, thereby elevating the polarization efficiency.

### 4.2. Ink-Based Moisture-Enabled Nanogenerators

Moisture-enabled nanogenerators (MEGs) offer an innovative approach to energy conversion by harnessing environmental moisture as a source of power, transforming the chemical energy contained within ubiquitous atmospheric humidity into electrical energy. Carbon-based materials play a pivotal role in this process by inducing the release of hydrogen ions from moisture, thereby facilitating moisture-induced electricity generation. Among these materials, GO stands out due to its excellent hydrophilicity, biocompatibility, and low cost, making it an ideal functional material for such applications. Despite the tendency of current research to enhance current output at the expense of voltage, this trade-off limits the potential of these devices for power supply applications. In an effort to address this challenge, Fandi Chen et al. [21] employed ultrasonic mixing techniques to incorporate varying ratios of rGO into GO suspensions, resulting in the fabrication of GO/rGO hybrid films with integrated GO and rGO heterostructures (as shown in Figure 21a). This hybrid film leverages the synergistic effects of the heterojunctions to expand the ion concentration gradient and reduce resistance, significantly optimizing the output voltage and current of GO/rGO MEGs (rGO:GO = 1:10), as illustrated in Figure 21b. Furthermore, Yulin Lv et al. [234] developed a high-performance paper-based electrokinetic generator by immersing commercial paper strips in carbon ink and coating one end with hygroscopic LiCl, achieving an output voltage of 0.35 V and current of 33.9 μA under conditions of 20 °C and 50% relative humidity. The addition of LiCl ensures spontaneous moisture absorption, and its asymmetric coating creates a directed capillary water flow of hydrated hydrogen ions within the paper-based electrokinetic generator. In pursuit of structural simplicity and ease of fabrication, Tiancheng He et al. [47] utilized screen printing techniques to create polyelectrolyte MEGs using poly-cation PDDA and poly-anion PSSA inks, incorporating active nano-Al on the PSSA side of the negative electrode to achieve high-performance output (Figure 21c). This design adopts a planar structure to maximize the interaction area with moisture and, by eliminating external metal relays and interconnectors, enables the fabrication of standardized power modules on a 10 cm × 20 cm PET substrate. Traditional evaporative-driven nanogenerators, which rely on the direct stacking of nanomaterials, face challenges related to material detachment and portability of the water tank. To overcome these issues, Xiaohan Zhao et al. [235] combined hydrophilic oxide NPs (e.g., silica/titania) with glass fiber support materials and a 3D network PVDF bonding material to form NPGF, which was then coated onto a flexible PET film substrate. Additionally, by employing a water storage sponge as a water supply device and printing MWCNT ink as electrodes, they fabricated an adhesive, stable, self-watering, evaporative-driven nanogenerator capable of withstanding repeated washing and deformation, as depicted in Figure 21d,e.

### 4.3. Ink-Based Hybrid Energy Nanogenerators

The advancement of hybrid energy harvesting technologies has substantially improved the overall power output of nanogenerators [239]. Current research in ink-based nanogenerators is primarily concentrated on the synergistic harvesting of solar thermal energy alongside triboelectric and TE mechanisms, in addition to the integration of TE and hygroelectric energy collection. Notably, there exists a research void concerning the integration of water–solid contact electrodes with solar cells and the in-depth exploration of various synergistic effects. Specifically, when water–solid contact TENGs are integrated with solar energy systems, they are capable of harnessing rainwater for continuous electricity generation during rainy seasons. In this vein, Busi Im et al. [238] have employed electrohydrodynamic jet printing to construct transparent conductive electrodes utilizing silver nanoparticles (Ag NPs), which were subsequently amalgamated with transparent hydrophobic PDMS and glass, culminating in the creation of a PA-TENG characterized by high transparency and elevated power output, as depicted in Figure 21h. By integrating this arrangement with solar cells, the efficiency of energy harvesting was significantly augmented. In comparison to ITO-TENGs, which utilize indium tin oxide (ITO) electrodes, the PA-TENG demonstrated a threefold increase in output power, attributable to the superior conductivity of Ag NP electrodes and the high contact potential barrier between Ag NP and PDMS. Moreover, Dan-Liang Wen et al. [237] fabricated a TEG with a serpentine arrangement of thermocouples by screen printing n-type (i.e., Bi${\;\;}_{2}$Te${\;\;}_{2.7}$Se${\;\;}_{0.3}$) and p-type (i.e., Sb${\;\;}_{2}$Te${\;\;}_{3}$) TE inks onto polyimide substrates. Incorporating a photothermal conversion layer, they engineered a hybrid photothermoelectric generator capable of concurrent absorption of thermal and radiant energies, as illustrated in Figure 21g. Additionally, Hygro-Thermo-Electricity, which is energy generated through the concerted action of moisture diffusion, the Soret effect, and the Seebeck effect in a moist thermal environment, was explored by Haoyu Shen et al. [236]. Through the synergistic use of bifunctional mobile ions and electrons, capitalizing on the Seebeck effect, the Soret effect, and the moisture-diffusion effect (as illustrated in Figure 21f), a TE substrate based on Bi${\;\;}_{2}$Te${\;\;}_{3}$/PEDOT:PSS was developed. This substrate, amenable to roll-to-roll printing, facilitates the modulation of the release of necessary ionic and electronic charges through glycerol absorption. This innovative flexible Hygro-Thermo-Electricity paper generator provides a groundbreaking method for exploiting low-grade hygrothermal resources in nature and for the development of future self-powered dwellings. Aerogels, characterized as nanostructured porous solid materials, exhibit outstanding thermal insulation capabilities due to their unique pore structures. The incorporation of CNTs within aerogels significantly enhances their electrical conductivity, paving the way for multifunctional material applications. In the context of the efficient utilization of marine energy resources, electromagnetic generators, TEG, and TENGs demonstrate considerable potential. Notably, K. Zhao et al. [88] have advanced this field by doping CNTs and aerogels into p-type and n-type Bi${\;\;}_{2}$Te${\;\;}_{3}$ TE materials, successfully fabricating high-performance TE films. Building on these findings, a high-performance hybrid energy harvesting prototype that integrates TEG, TENG, and electromagnetic generator has been designed and realized. This work not only optimizes the thermal-to-electrical conversion efficiency but also charts a new course for the efficient and sustainable exploitation of marine energy resources.

## 5. The Latest Advances in Ink-Based Nanogenerator Applications

The advent of printing technology in the fabrication of nanogenerators has ushered in a new era of high-performance ink-based nanogenerators, which exhibit significant potential as energy supply devices for compact equipment. These nanogenerators are adept at transforming ambient physical energy into electrical energy, positioning them as promising candidates for autonomous sensing devices. This manuscript will not differentiate among the five previously mentioned distinct types of ink-based nanogenerators. Instead, it will offer a comprehensive overview of their latest advancements in the realm of energy provisioning and autonomous sensing capabilities, maintaining a focus on their application as innovative power solutions.

### 5.1. Power Supply Devices

The current paradigm in consumer electronics is progressively evolving towards wearable technologies, necessitating more adaptable and integrated power solutions. Conventional batteries, burdened by their substantial weight and considerable size, fall short of fulfilling the demands for distributed energy provisioning in such applications. Nanogenerator technologies, epitomized by TEGs, TENGs, PENGs, and moisture-driven nanogenerators, stand out as the most promising avenues for powering future wearable electronics. There has been notable progress in the development of materials and structural designs for ink-based nanogenerators. This discourse aims to explore the recent advancements in the application of these four categories of ink-based nanogenerators as power sources, underscoring their transformative potential in the realm of wearable electronics.

To exploit natural low-grade humid thermal resources, Haoyu Shen et al. [236] developed a flexible Hygro-Thermo-Electricity paper generator and envisioned a self-powered house comprising multiple brick-shaped generators (Figure 22a). Yan Wang and colleagues [38] crafted a high-performance UF-TENG using a dual-layer conductive ink film composed of PET and polytetrafluoroethylene, further designing a prototype for a self-powered underwater buoy, as depicted in Figure 22b. A cylindrical structure was utilized to induce Karman vortex street enhancement for UF-TENG vibrations. Inspired by LEGO bricks, Fang Chen et al. [46] fabricated a PENG aimed at harvesting energy from jumping motions. This was accompanied by a schematic of a multi-material human shoe insole designed for energy collection, illustrated in Figure 22c. Bhavna Hedau and team [34] synthesized a self-polarized MoS${\;\;}_{2}$@PVDF hybrid nanocomposite film with an enhanced triboelectric effect, achieved through solution processing coupled with rod material printing techniques. The resultant high-performance TENGs delivered a V${\;\;}_{OC}$ of 72 V and an I${\;\;}_{SC}$ of 4.4 μA, with its application for charging smartwatches showcased in Figure 22d. Tiancheng He et al. [47] employed silk-screen printing to produce a fully printed planar magnetoencephalography (PMEG) array using polycationic PDDA and polyanionic PSSA inks, achieving an impressive V${\;\;}_{OC}$ of up to 1.1 V and a power density of 2.6 mW cm${\;\;}^{-2}$, demonstrating its utility in charging electronic watches (Figure 22e). Moreover, a summary of several performance metrics of ink-based TENGs as energy-supplying devices is presented in Table 2.

### 5.2. Self-Powered Sensing Devices

Self-powered sensing applications of ink-based nanogenerators are emerging as a pivotal advancement within the wearable electronics domain. Zhenwei Wang et al. [228] have adeptly fabricated a customizable, fully flexible single-electrode TENG, showcasing its utility as a self-powered tactile sensor (refer to Figure 23a). Furthering this innovation, Wenxiu Wu and collaborators [179] utilized high-conductivity ink composed of short Ag NWs to develop a TENG for imbalance detection in robotic systems (refer to Figure 23b). In a notable endeavor, Ping Sun and co-researchers [240] leveraged heat-resistant silicone as a fire-resistant and heat-resistant triboelectric negative electrode material, engineering a self-powered multi-arch E-TENG bending sensor and G-TENG pressure sensor. Integration with Support Vector Machine algorithms facilitated the precise perception of human motions such as gestures and gaits in high-temperature conditions (refer to Figure 23c). Shipeng Zhang et al. [39] employed 3D printing technology to construct a self-powered, single-electrode, simplistically structured ring-shaped Triboelectric Sensor, illustrating its potential for human–machine interaction applications (refer to Figure 23d). The innovative transparent self-healing 4D printed TENG developed by Long-Biao Huang and his team [182] demonstrated proficiency in monitoring human joint bending angles within self-powered sensor systems (refer to Figure 23e). Yunqing He and associates [25] devised a paper-based TEG, proving its efficacy in motion and breath detection applications (refer to Figure 23f). Xingzhong Zhang and his team [167] showcased the creation of a high-performance paper-based TEG via screen printing technology, eliminating the need for hot-press sintering. This breakthrough revealed the application of a programmable TE infrared dynamic display equipped with optical encryption and anti-counterfeiting capabilities (refer to Figure 23g), underscoring the promising potential of paper substrates as alternatives to conventional polymer bases.

## 6. Challenges, Outlook, and Future Perspectives of Ink-Based Nanogenerators

In conclusion, ink-based printing technology has forged new avenues for the selection of materials and the design of structures in nanogenerators. This comprehensive review sheds light on the recent advancements in material optimization and structural design, specifically within ink-based TEGs and TENGs. Furthermore, it delves into the progress of ink-based PENGs, MEGs, and hybrid energy nanogenerators. Notably, the discussion begins by highlighting the crucial optimizations made to TE material inks, encompassing metallic-based, polymer composite, carbon-based materials, and CQDs. It subsequently details the innovative designs of ink-based TEGs in planar, 3D, and bulk configurations, alongside advancements in substrate protection and flexibility enhancement. Moreover, the review examines the development of triboelectric ink materials, including conductive, positive, and negative dielectric inks, and inks with unique functionalities and introduces TENGs featuring micro-surface, linear, protruding, and bulk structures, as well as pulse output configurations. Additionally, the review extends to ink-based PENGs, MEGs, and hybrid energy nanogenerators. When compared to traditional manufacturing methods, ink-based printing technology unveils the potential for the commercial application of nanogenerators.

Despite the notable progress achieved in these ink-based nanogenerators, each domain still confronts its own set of challenges:In the field of ink-based TEGs, significant progress has been made in the long-term stability and good dispersibility of TE inks, but improving the TE performance of materials remains a top priority, especially for n-type organic TE materials. The current performance enhancement of organic ink materials is mostly achieved through the doping of toxic inorganic precious metal NPs. Among these, carbon-based nanocomposite materials and CQD materials show great potential. In terms of structural design, further exploration of efficient 3D heat-collecting structures is warranted, along with the continued enrichment of substrate optimization and protection methods.In the field of ink-based TENGs, the electrical performance of conductive inks and dielectric inks should be further improved, and efforts should be made to find corrosion-resistant ink materials or designs that can adapt to complex environments. Special emphasis should be placed on improving the electrical performance of n-type polymer dielectric inks and exploring methods that minimize or eliminate the use of inorganic precious metal NPs. The relative motion of functional layers can also accelerate ink wear to some extent; thus, exploring the potential of self-healing inks is necessary. Structural design can continue to enrich the research on linear micro-surfaces, protruding micro-surfaces, block structures, and even biomimetic microstructures.In the field of ink-based PENGs, the active exploration of performance enhancement of PE ink materials through the enrichment of solution-induced β-phase PVDF is needed. Moreover, optimizing the stacked structural design remains an effective way to significantly improve performance and deserves active exploration.For ink-based MEGs, research is relatively weak, with most studies focusing on carbon-based ink materials. It is necessary to actively expand the range of ink material choices and optimize ink materials. Additionally, exploring efficient water vapor absorption structural designs for energy conversion is essential.For ink-based hybrid energy nanogenerators, the active exploration of the synergistic principles and methods of multiple energy conversion effects is required. Additionally, optimizing integration and efficient connection under multiple power generation mechanisms can significantly enhance output performance.

In summary, ink-based nanogenerators are steadily advancing towards commercialization, with substantial progress in the development of functional inks and structural designs over the last three years. Interdisciplinary collaboration is anticipated to propel continuous advancements in nanogenerator technology. It is confidently believed that, through relentless optimization and innovation, ink-based nanogenerators will increasingly meet the diverse requirements of future flexible and wearable electronic devices and sensors.

## Figures and Tables

**Figure 1 ijms-25-06152-f001:**
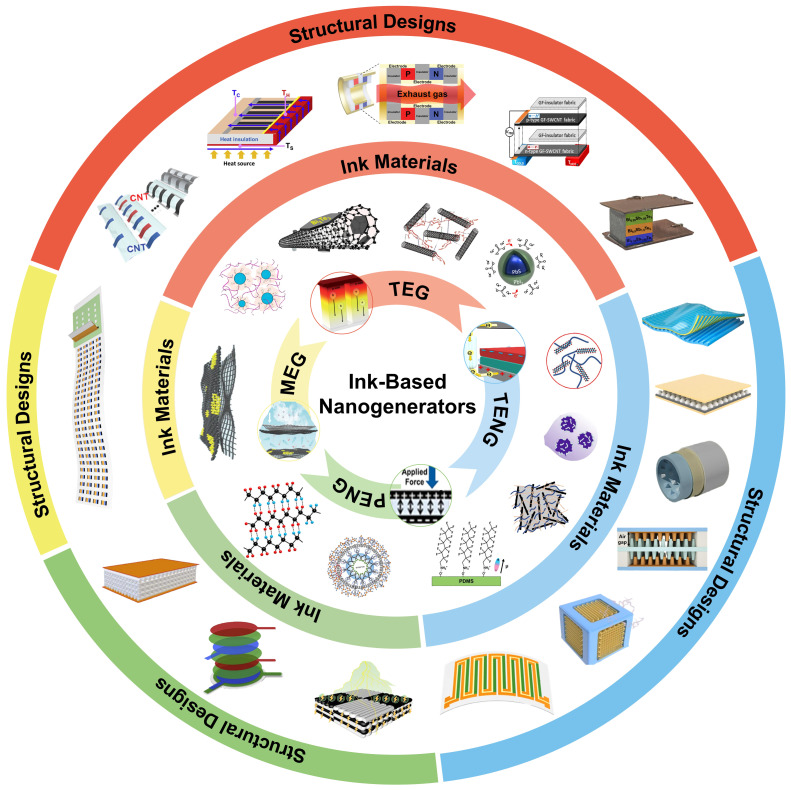
Overview of the latest developments in ink-based nanogenerators. Reprinted with permission from [18,19,20,21,22,23,24,25,26,27,28,29,30,31,32,33,34,35,36,37,38,39,40,41,42,43,44,45,46,47].

**Figure 2 ijms-25-06152-f002:**
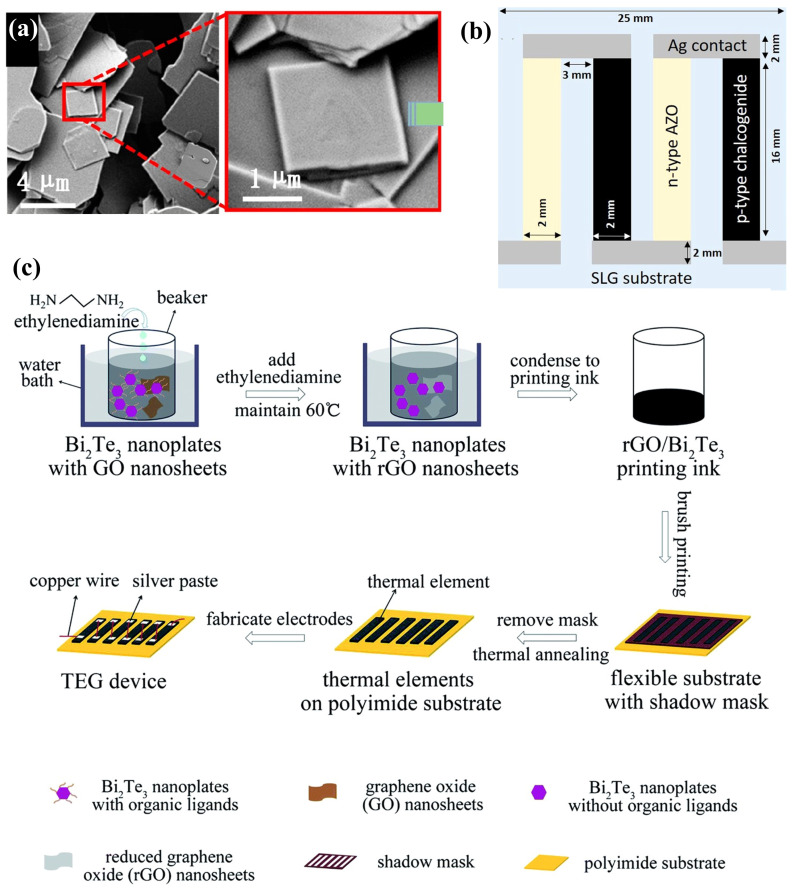
Solution chemical synthesis of inorganic TE inks: microstructural characterization, ink preparation, and device fabrication. (**a**) SEM micrographs of SnSe nanoplates. Reprinted with permission from [57]. Copyright 2020, American Chemical Society. (**b**) Schematic representation of a chalcogenide/AZO thin-film TEG, not drawn to scale. Reprinted with permission from [78]. Copyright 2022, American Chemical Society. (**c**) Schematic of the fabrication process of rGo/Bi${\;\;}_{2}$Te${\;\;}_{3}$ ink mixture. Reprinted with permission from [56]. Copyright 2018, Royal Society of Chemistry.

**Figure 3 ijms-25-06152-f003:**
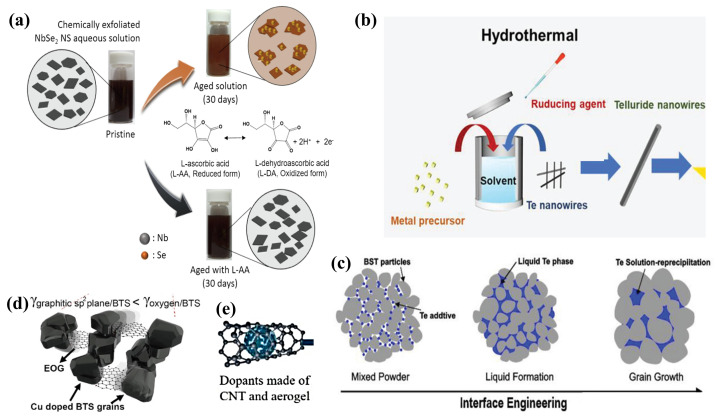
Preparation of stable ink and schematic of nanoparticle interface connection. (**a**) Schematic illustration of the preparation process for stable NSs−based TE ink under the redox action of L−AA. Reprinted with permission from [74]. Copyright 2023, Elsevier. (**b**) Schematic illustration of the preparation of Te NWs. Reprinted with permission from [81]. Copyright 2020, John Wiley and Sons. (**c**) Schematic illustration of the evolution of Te−based nanosolder bridging at the interface of BST particles. Reprinted with permission from [86]. Copyright 2019, John Wiley and Sons. (**d**) Schematic illustration of edge−oxidized graphene bridging at the interface of Cu−doped BST grain boundaries. Reprinted with permission from [87]. Copyright 2021, American Chemical Society. (**e**) Schematic of interface connection with CNT and aerogel doping. Reprinted with permission from [88]. Copyright 2023, Elsevier.

**Figure 4 ijms-25-06152-f004:**
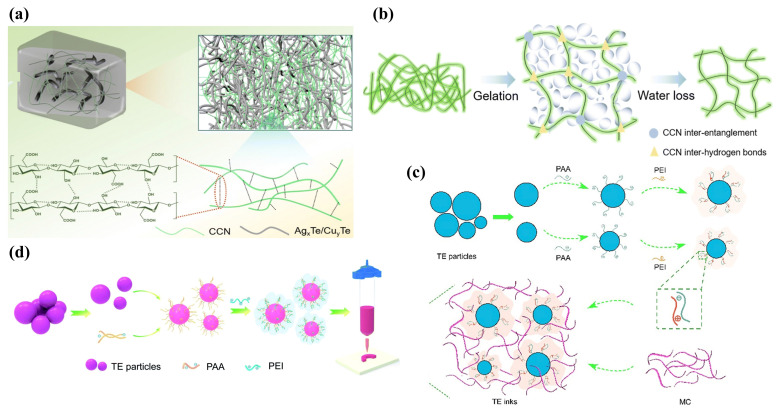
Mechanism of improvement in viscoelasticity of inks. (**a**) Schematic of hydrogen bond restriction between CCNs and dispersion of Ag${\;\;}_{x}$Te/Cu${\;\;}_{y}$Te NRs. Reprinted with permission from [99]. Copyright 2009, Royal Society of Chemistry. (**b**) Schematic diagram of inter-entanglement and physical crosslinking in CCN hydrogel. Reprinted with permission from [99]. Copyright 2009, Royal Society of Chemistry. (**c**) Schematic of TE particles induced by PAA and PEI. Reprinted with permission from [23]. Copyright 2023, Elsevier. (**d**) Schematic illustration of the stable hybrid structure formed by PAA and PEI with Bi${\;\;}_{2}$Te${\;\;}_{3}$-based TE particles. Reprinted with permission from [100]. Copyright 2022, Royal Society of Chemistry.

**Figure 5 ijms-25-06152-f005:**
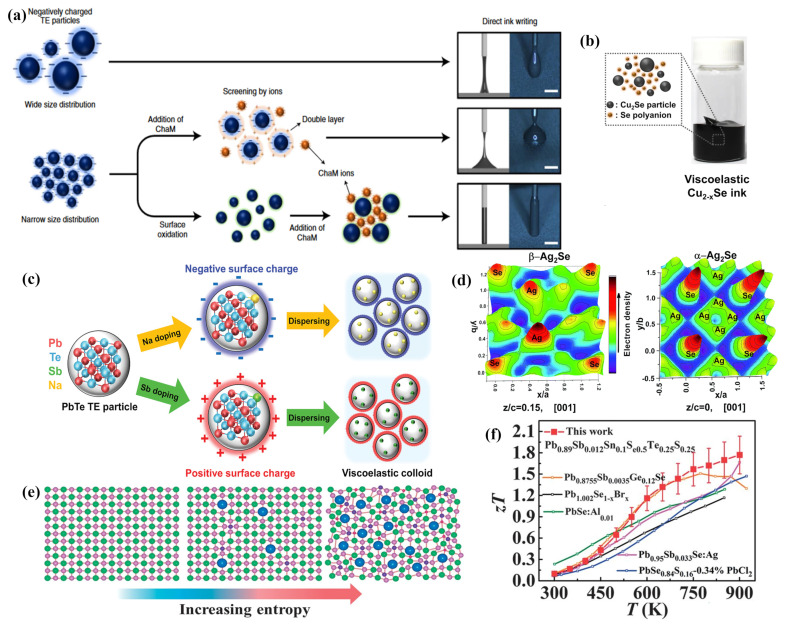
Application of inorganic improvement methods in the preparation of elastic TE inks, from surface treatment to lattice optimization. (**a**) Design of surface oxidation and ChaM addition for super−elastic TE Inks. Reprinted with permission from [96]. Copyright 2021, Springer Nature. (**b**) Schematic of Cu${\;\;}_{2}$Se−based ink with viscoelasticity optimized by the electrovisection effect of Se${\;\;}_{8}^{2-}$poly−anions. Reprinted with permission from [103]. Copyright 2021, Springer Nature. (**c**) Schematic illustration of viscoelasticity optimization through surface charge on PbTe particles induced by inorganic Se${\;\;}_{8}^{2-}$poly−anions. Reprinted with permission from [28]. Copyright 2021, John Wiley and Sons. (**d**) The volatilization of Se from Ag particles, with low−temperature annealing forming the orthorhombic β−Ag${\;\;}_{2}$Se TE phase. Reprinted with permission from [67]. Copyright 2013, Royal Society of Chemistry. (**e**) Schematic of lattice distortion in PbSe with increasing entropy following the introduction of Sn. The pink, red, green, blue, and purple spheres represent Pb, Sn, Se, Te, and S atoms, respectively. Reprinted with permission from [104]. Copyright 2021, Science. (**f**) Temperature dependence of ZT values. Reprinted with permission from [104]. Copyright 2021, Science.

**Figure 6 ijms-25-06152-f006:**
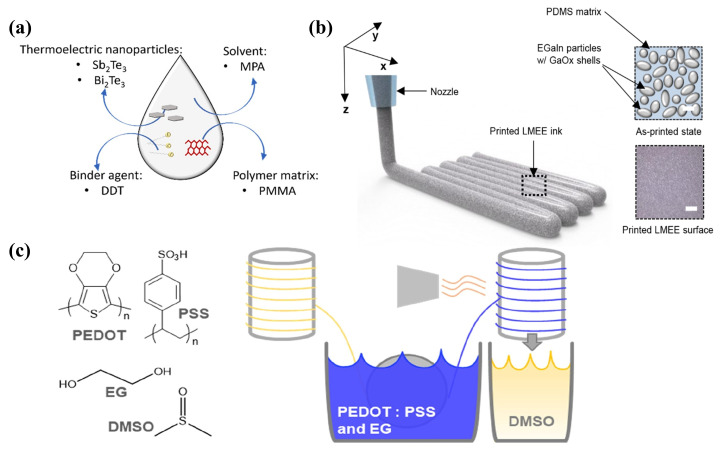
Schematic illustration of polymer ink composition. (**a**) PMMA TE ink embedding hexagonal flake-like Sb${\;\;}_{2}$Te${\;\;}_{3}$ and Bi${\;\;}_{2}$Te${\;\;}_{3}$. Reprinted with permission from [119]. Copyright 2022, American Chemical Society. (**b**) LMEE ink based on EGaIn-PDMS emulsion. Reprinted with permission from [120] Copyright 2022, American Chemical Society. (**c**) Schematic of high conductivity yarn preparation based on PEDOT:PSS conductive ink. Reprinted with permission from [121]. Copyright 2020, American Chemical Society.

**Figure 7 ijms-25-06152-f007:**
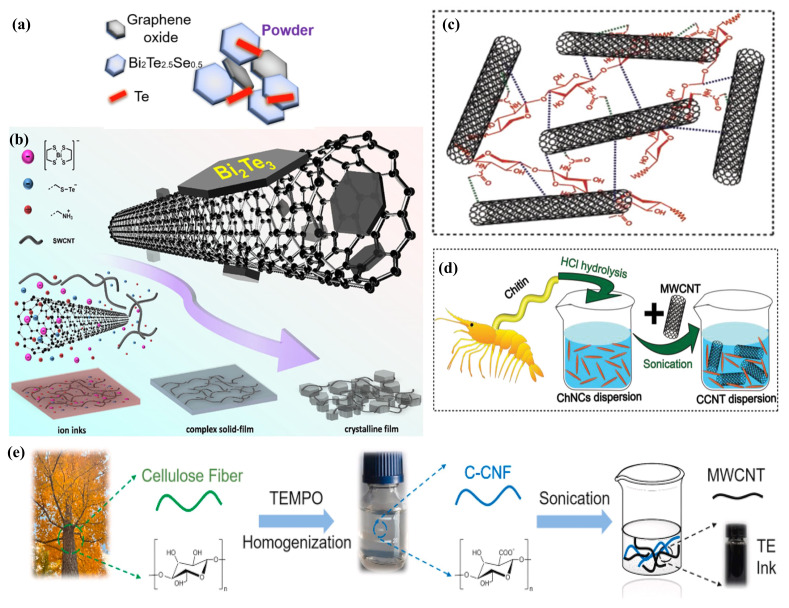
Interfacial engineering and ink preparation process. (**a**–**c**): Schematic of interfacial engineering. (**a**) Schematic of interfacial engineering for Bi${\;\;}_{2}$Te${\;\;}_{2.5}$Se${\;\;}_{0.5}$-GO. Reprinted with permission from [130]. Copyright 2022, Chemical Engineering Journal. (**b**) Cement (Bi${\;\;}_{2}$Te${\;\;}_{3}$ nanolayer)–rebar (SWCNT) interfacial engineering schematic. Reprinted with permission from [24]. Copyright 2021, John Wiley and Sons. (**c**) Schematic of interfacial engineering between ChNCs and CNTs. Reprinted with permission from [25]. Copyright 2022, John Wiley and Sons. (**d**,**e**): ink preparation process. (**d**) Ultrasonic treatment process for preparing ChNCs/MWCNT (CCNT) ink. Reprinted with permission from [25]. Copyright 2023, Elsevier. (**e**) Fabrication process for p-type TE ink based on MWCNT and C-CNF. Reprinted with permission from [106]. Copyright 2021, Elsevier.

**Figure 8 ijms-25-06152-f008:**
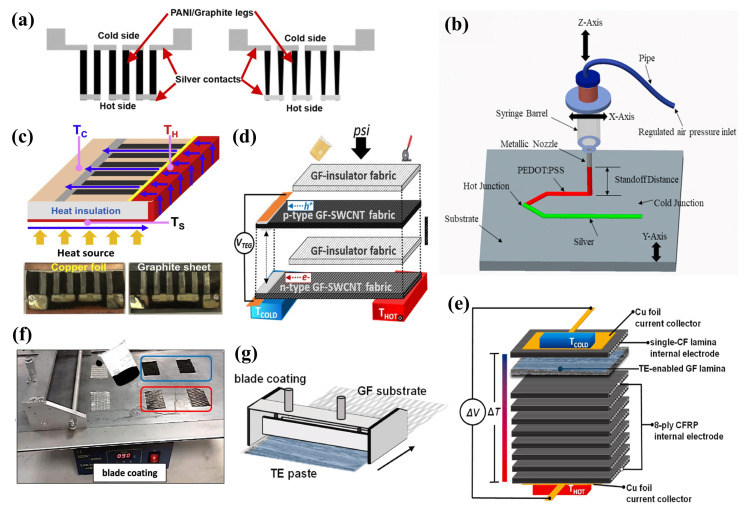
Schematic of planar and stacked structures that can enhance TEG performance. (**a**–**c**): schematic of the planar structure. (**a**) Comparison of rectangular and trapezoidal TEG legs. Reprinted with permission from [151]. Copyright 2022, Elsevier. (**b**) Comparison of TEGs with different characteristic lengths of 30 and 40 mm. Reprinted with permission from [48]. Copyright 2021, Elsevier. (**c**) Directional heat collection design combining HTL and insulation foam. Reprinted with permission from [26]. Copyright 2020, Elsevier. (**d**) Schematic illustration of the 16-layer structural TEG configuration based on glass-fiber-reinforced polymer composite laminates. Reprinted with permission from [27]. Copyright 2021, American Chemical Society. (**e**,**g**) Lamination material fabrication process. (**e**) Schematic of the 8-layer TEG structure based on UD laminate. Reprinted with permission from [152]. Copyright 2022, Elsevier. (**f**) Fabrication process of GF-SWCNT layers. Reprinted with permission from [27]. Copyright 2021, American Chemical Society. (**g**) Fabrication process of GF-SWCNT fabric. Reprinted with permission from [152]. Copyright 2022, Elsevier.

**Figure 9 ijms-25-06152-f009:**
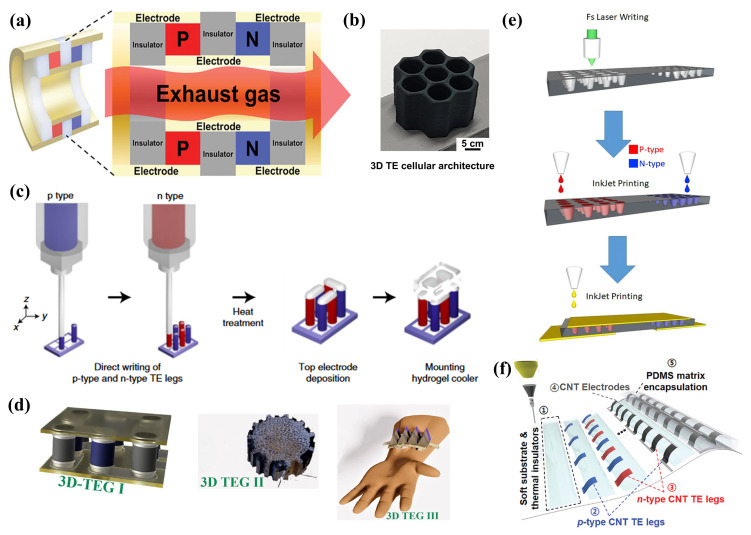
Schematic illustration of a 3D TEG structure. (**a**) Custom tubular TEG structure for waste heat recovery. Reprinted with permission from [28]. Copyright 2021, John Wiley and Sons. (**b**) Efficient honeycomb TEG structure with mechanical rigidity. Reprinted with permission from [103]. Copyright 2021, Springer Nature. (**c**) 3D arch-shaped micro TEG structure. Reprinted with permission from [96]. Copyright 2021, Springer Nature. (**d**) Rectangular-shaped TEG-I structure, cylindrical gear-shaped TEG-II structure, sawtooth-shaped TEG-III structure. Reprinted with permission from [158]. Copyright 2021, American Chemical Society. (**e**) Thermoelectric leg-embedded PEN foil TEG structure. Reprinted with permission from [29]. Copyright 2020, Elsevier. (**f**) Arch-shaped TEG structure. Reprinted with permission from [30]. Copyright 2023, John Wiley and Sons.

**Figure 10 ijms-25-06152-f010:**
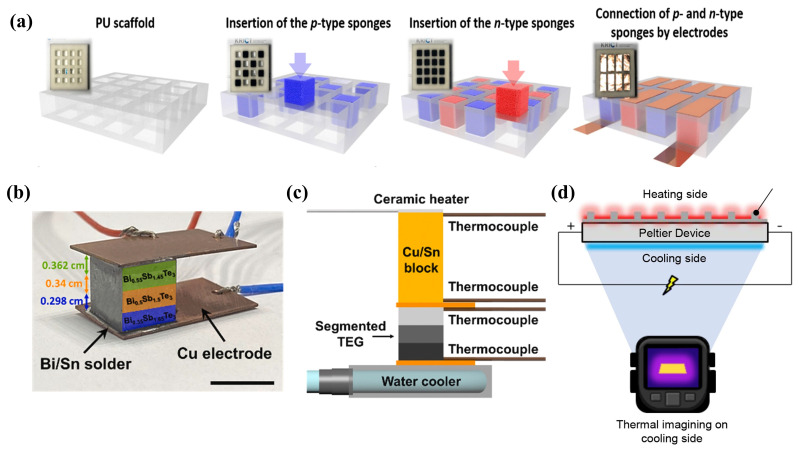
Schematic illustration of bulk TEG structure. (**a**) Vertical structure sponge TEG. Reprinted with permission from [160]. Copyright 2020, Elsevier. (**b**) Three-dimensionally printed sequentially deposited three-layer Bi${\;\;}_{x}$Sb${\;\;}_{2-x}$Te${\;\;}_{3}$ TE ink. Reprinted with permission from [31]. Copyright 2021, Elsevier. (**c**) Schematic of the overall structure of a segmented TEG. Reprinted with permission from [31]. Copyright 2021, Elsevier. (**d**) TEG based on LMEE structure. Reprinted with permission from [120]. Copyright 2022, American Chemical Society.

**Figure 11 ijms-25-06152-f011:**
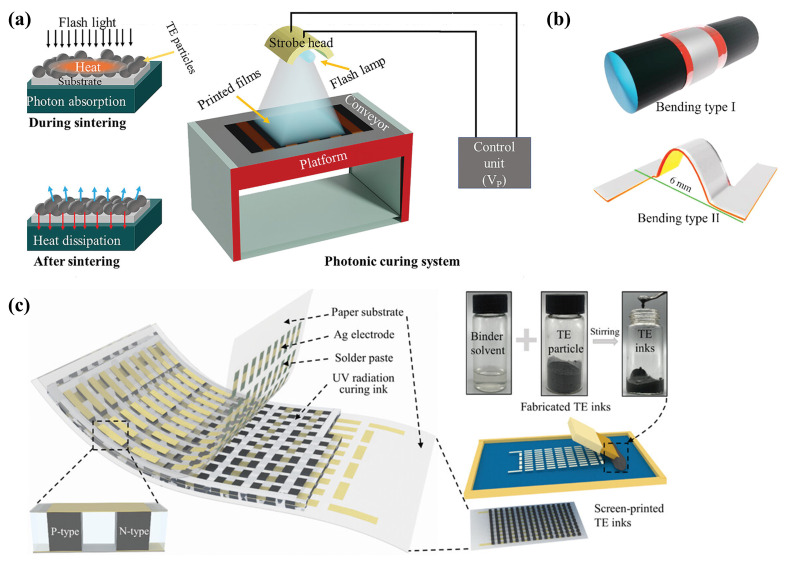
Application of advanced substrate protection techniques in the preparation of flexible TEGs: from micro-welding to tearable paper-based designs. (**a**) Millisecond photonic curing grain “micro-welding” process. Reprinted with permission from [166]. Copyright 2022, John Wiley and Sons. (**b**) Millimeter-thick flexible p-BST and n-BT TE films. Reprinted with permission from [166]. Copyright 2023, Elsevier. (**c**) Schematic of tearable paper-based TEG without the need for hot-press sintering and complex fabrication processes. Reprinted with permission from [167]. Copyright 2022, John Wiley and Sons.

**Figure 12 ijms-25-06152-f012:**
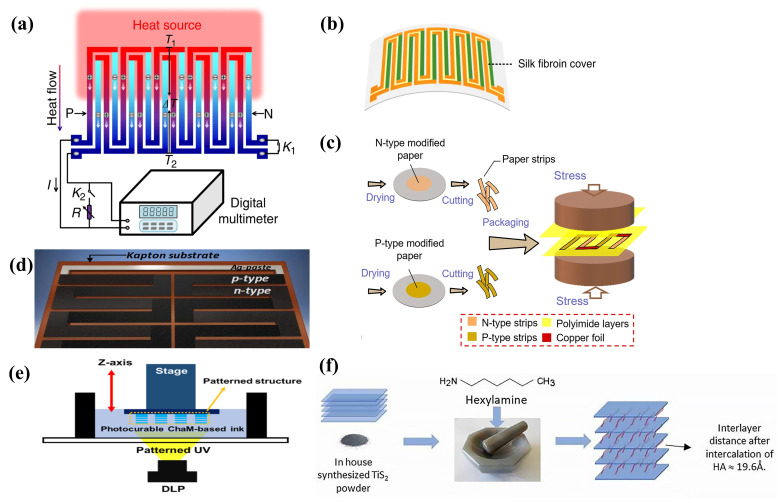
Flexible structure design and innovative preparation process of ink-based TEGs. (**a**) Schematic of the dual-chain thermoelectric generator (DC-ThEG) structure. Reprinted with permission from [41]. Copyright 2020, Springer Nature. (**b**) Silk fibroin material covering the gap between chains as sensing electrodes. Reprinted with permission from [41]. Copyright 2020, Springer Nature. (**c**) Modification process of paper-based TEG preparation. Reprinted with permission from [168]. Copyright 2021, Springer Nature. (**d**) Schematic of an all-carbon organic TEG structure without metal junctions. Reprinted with permission from [138]. Copyright 2021, American Chemical Society. (**e**) Optical printing process based on Digital Light Processing. Reprinted with permission from [169]. Copyright 2022, Springer Nature. (**f**) Improved process for electrochemical intercalation of TiS${\;\;}_{2}$/HA. Reprinted with permission from [170]. Copyright 2021, Taylor & Francis.

**Figure 13 ijms-25-06152-f013:**
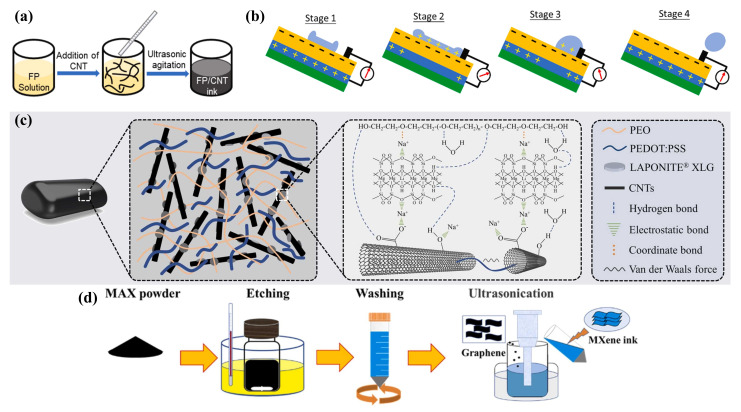
(**a**,**d**): The preparation process of carbon matrix composite inks. (**a**) Schematic of the synthesis process for FP/CNT ink. Reprinted with permission from [176]. Copyright 2021, Elsevier. (**b**) Droplet−based electricity generator power generation mechanism. Reprinted with permission from [176]. Copyright 2021, Elsevier. (**c**) LPPC ink composition and interfacial connection schematic. Reprinted with permission from [32]. Copyright 2023, Elsevier. (**d**) Schematic diagram of the preparation of MXene/graphene ink. Reprinted with permission from [177]. Copyright 2022, Elsevier.

**Figure 15 ijms-25-06152-f015:**
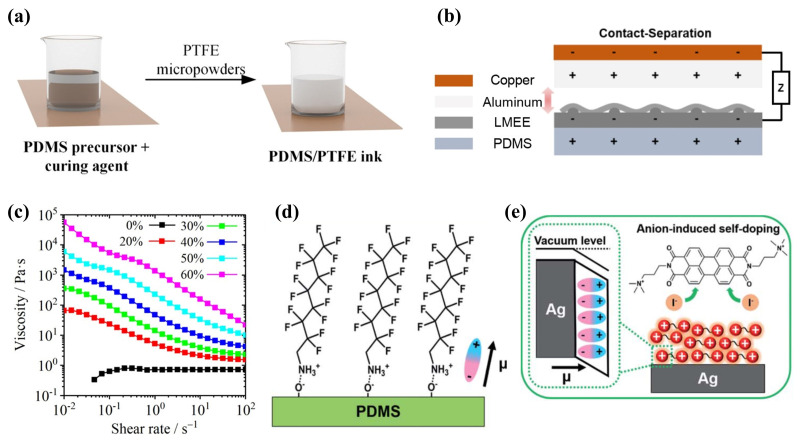
Development of special functional inks for ink−based TENGs. (**a**) Schematic illustration of the preparation of PDMS/PTFE ink. Reprinted with permission from [202]. Copyright 2020, American Chemical Society. (**b**) Detailed structure of TENG with LMEE structure for enhanced charge transfer. Reprinted with permission from [120]. Copyright 2022, American Chemical Society. (**c**) The effect of different PTFE contents on the viscosity of inks when the ratio of PDMS to curing agent is 1:3. Reprinted with permission from [202]. Copyright 2020, American Chemical Society. (**d**) PDMS with F${\;\;}_{15}-$NH${\;\;}_{2}$ surface modification. Reprinted with permission from [35]. Copyright 2013, Royal Society of Chemistry. (**e**) PDMS with F${\;\;}_{15}-$NH${\;\;}_{2}$ surface modification. Reprinted with permission from [35]. Copyright 2013, Royal Society of Chemistry.

**Figure 16 ijms-25-06152-f016:**
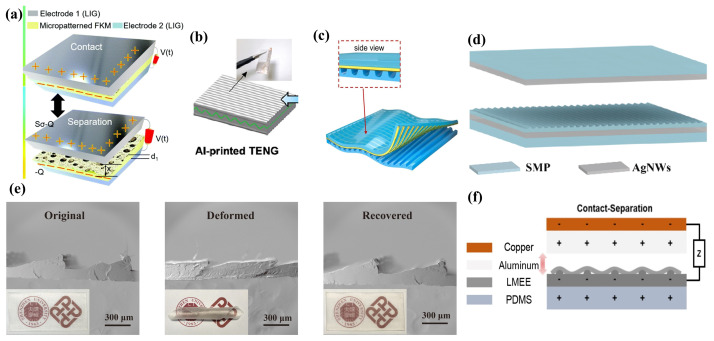
Planar microstructure and linear microstructure of ink−based TENG. (**a**) The FKM frictional electronegative layer with a porous microstructure. Reprinted with permission from [180]. Copyright 2022, Royal Society of Chemistry. (**b**) Schematic diagram of silicone layer protected wrinkled silver electrode. Reprinted with permission from [181]. Copyright 2020, Elsevier. (**c**) Schematic diagram of microporous Ag NWs line array structure. Reprinted with permission from [37]. Copyright 2023, Elsevier. (**d**) Schematic diagram of SMP negative charge layer TENG structure with wire−like surface microstructure. Reprinted with permission from [182]. Copyright 2021, Elsevier. (**e**) Stimulation and recovery process of the shape memory material SMP. Reprinted with permission from [182]. Copyright 2021, Elsevier. (**f**) Schematic diagram of TENG structure with LMEE configuration enhancing charge transfer of the frictional electronegative layer. Reprinted with permission from [120]. Copyright 2022, American Chemical Society.

**Figure 17 ijms-25-06152-f017:**
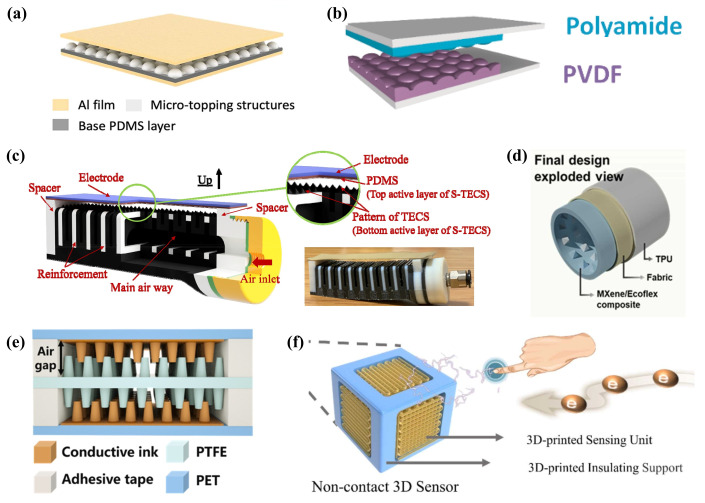
Schematic diagram of a 3D ink-based TENG structure, including protruding structures, bulk structures, and parts with special structures. (**a**,**b**) are hemispherical structures. (**a**) Detailed schematic diagram of the micro-top-layer structure TENGs structure. Reprinted with permission from [36]. Copyright 2023, Elsevier. (**b**) Schematic diagram of TENG structure with interlocking hemispherical array on the surface. Reprinted with permission from [210]. Copyright 2021, Elsevier. (**c**–**e**) are pyramid-like structures. (**c**) Self-powered triboelectric curvature sensor. Reprinted with permission from [211]. Copyright 2020, Elsevier. (**d**) Structure of the self-powered toroidal triboelectric sensor. Reprinted with permission from [39]. Copyright 2023, Elsevier. (**e**) Cylindrical support structure with UF-TENG in flowing water. Reprinted with permission from [38]. Copyright 2021, Elsevier. (**f**) Proximity Non-Contact TENG with a bulk structure. Reprinted with permission from [40]. Copyright 2023, Elsevier.

**Figure 18 ijms-25-06152-f018:**
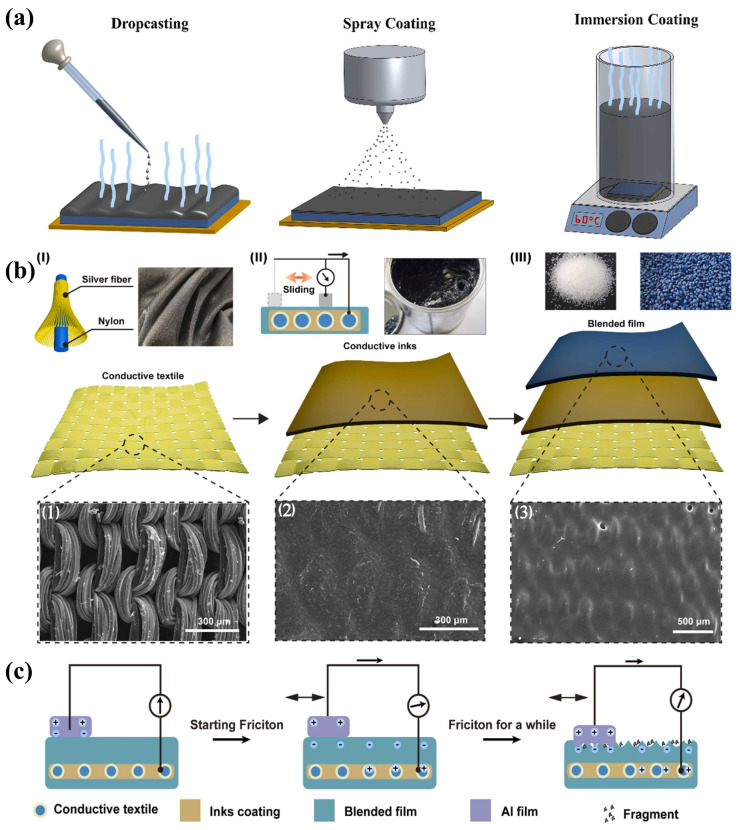
(**a**) Preparation of graphene electrodes using drop casting, spray coating, and immersion methods. Reprinted with permission from [220]. Copyright 2023, Elsevier. (**b**) Layered the conductive fibers in flexible textile organic structure and SEM schematic of TENG. (**I**) The 3D structure of conductive fabric, with illustrations including actual photographs of the conductive fabric and the basic structure of conductive fibers. (**II**) The 3D structure of conductive fabric coated with conductive ink, with illustrations depicting the operational cycle of OTG and actual photographs of the conductive ink. (**III**) The 3D structure of a blended film, with illustrations showing actual photographs of PVA and PEDOT. (**1**–**3**) SEM images of conductive textile and the surface appearance of hydrophilic conductive ink, along with surface profiles of PVA and PEDOT blended films. Reprinted with permission from [221]. Copyright 2023, Elsevier. (**c**) Schematic of the electricity generation cycle and working mechanism of flexible textile organic TENG. Reprinted with permission from [221]. Copyright 2023, Elsevier.

**Figure 19 ijms-25-06152-f019:**
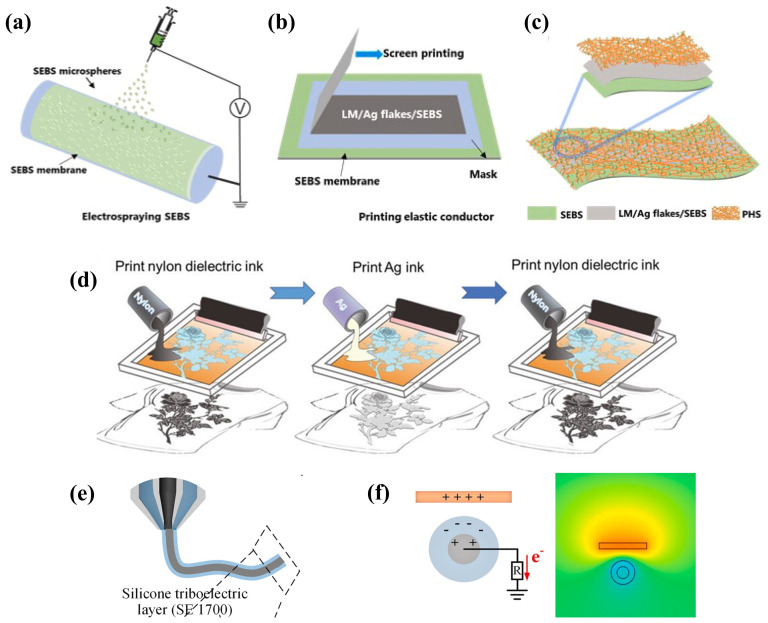
Preparation process schematic. (**a**) Simultaneous electrospraying of SEBS and electrospinning of PVDF−HFP for the preparation of PHS technology. Reprinted with permission from [226]. Copyright 2020, Elsevier. (**b**) Screen printing of LM/Ag electrodes. Reprinted with permission from [226]. Copyright 2020, Elsevier. (**c**) Self−interlocked stretchable nanofiber based TENG structure. Reprinted with permission from [226]. Copyright 2020, Elsevier. (**d**) Alternating printing of nylon ink and silver ink to fabricate custom−patterned screen−printed textile TENG. Reprinted with permission from [227]. Copyright 2022, John Wiley and Sons. (**e**) Schematic diagram of the coaxial printing process. Reprinted with permission from [228]. Copyright 2021, Elsevier. (**f**) One working state of the single−electrode FFTENG and its COMSOL simulation. Reprinted with permission from [228]. Copyright 2021, Elsevier.

**Figure 20 ijms-25-06152-f020:**
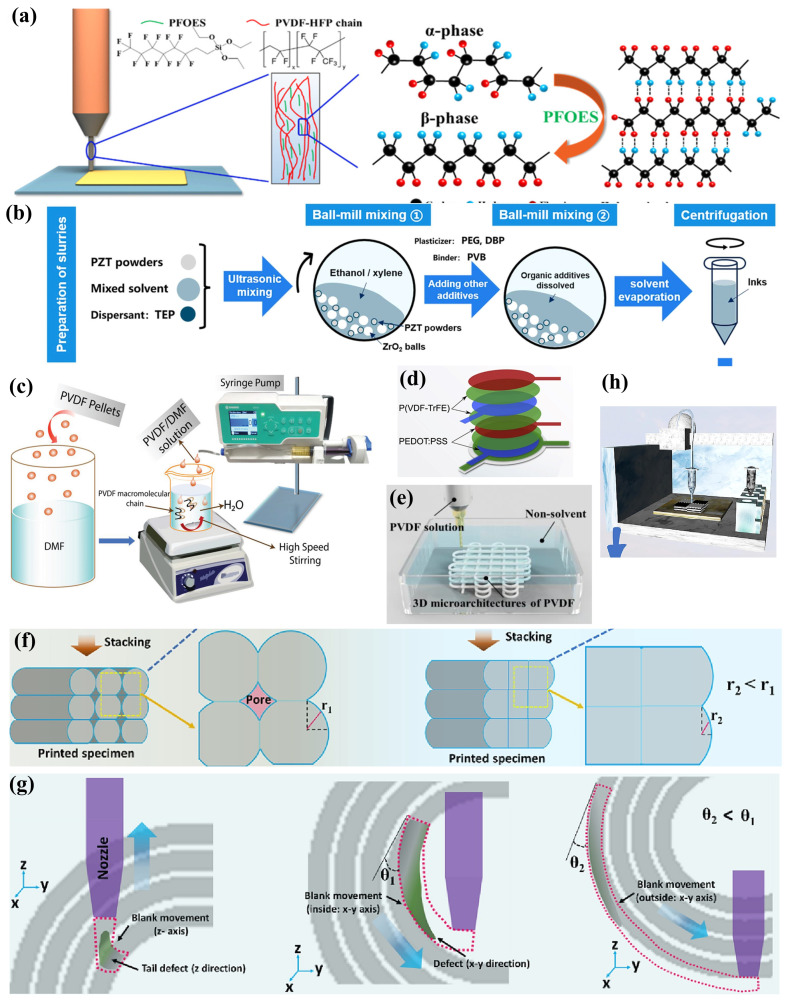
(**a**–**c**): PE ink optimization. (**a**) Schematic illustration of the phase transition of PVDF-HFP to the β-phase under the influence of PFOES. Reprinted with permission from [42]. Copyright 2009, Royal Society of Chemistry. (**b**) Preparation process of DIW suspension of organic-based PZT powder. Reprinted with permission from [44]. Copyright 2022, Elsevier. (**c**) Solution treatment phase separation technique for the preparation of δ-phase PVDF NPs. Reprinted with permission from [43]. Copyright 2022, John Wiley and Sons. (**d**–**h**) are performance enhancements through process optimization of PENGs. (**d**) Schematic of 10-layer all-printed P(VDF-TrFE) PE polymer. Reprinted with permission from [45]. Copyright 2020, Elsevier. (**e**) Schematic of solvent-assisted direct-write 3D printing process. Reprinted with permission from [232]. Copyright 2022, Elsevier. (**f**) Inter-strut void generation. Reprinted with permission from [233]. Copyright 2023, Elsevier. (**g**) Tail defect. Reprinted with permission from [233]. Copyright 2023, Elsevier. (**h**) P-layered alternating printing for the preparation of voxel-isolated nanogenerators. Reprinted with permission from [46]. Copyright 2023, Elsevier.

**Figure 21 ijms-25-06152-f021:**
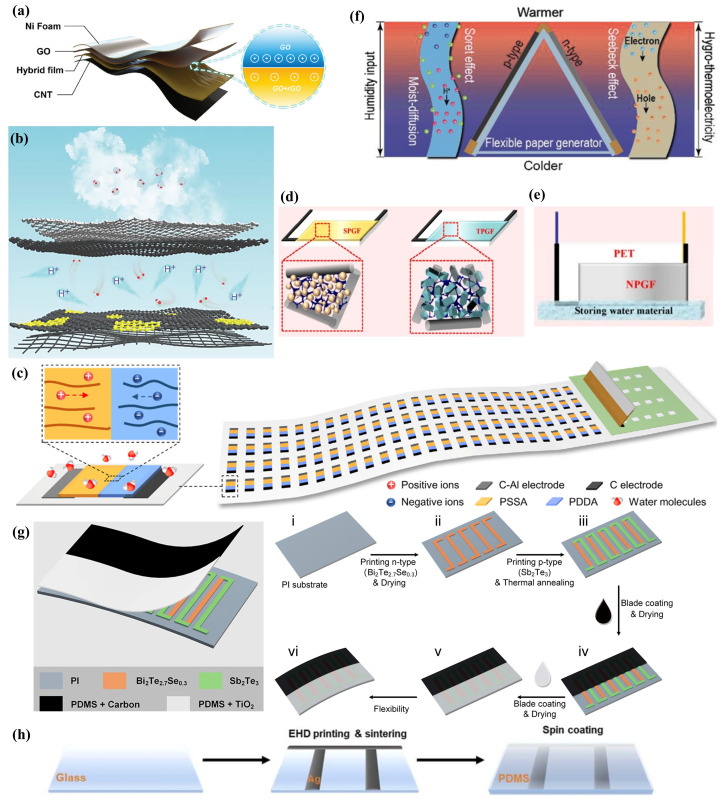
MEG and hybrid energy nanogenerator mechanism, structure and preparation process. (**a**) Schematic illustration of the GO/rGO hybrid film structure. Reprinted with permission from [21]. Copyright 2023, John Wiley and Sons. (**b**) GO/rGO MEGs power generation mechanism. Reprinted with permission from [21]. Copyright 2023, John Wiley and Sons. (**c**) PMEG power generation mechanism and screen printing manufacturing process schematic. Reprinted with permission from [47]. Copyright 2023, Elsevier. (**d**) SPGF/TPGF composition. Reprinted with permission from [235]. Copyright 2023, Elsevier. (**e**) Schematic of the self-water-supplied nanogenerator structure driven by water evaporation. Reprinted with permission from [235]. Copyright 2022, Elsevier. (**f**) “Hygro-TE” power generation mechanism that synergistically utilizes the Seebeck effect, Soret effect, and moisture-diffusion effect. Reprinted with permission from [236]. Copyright 2023, John Wiley and Sons. (**g**) Schematic illustration of the preparation of the hybrid photo thermoelectric generator. (**i**,**ii**) The n-type thermoelectric ink was applied onto a flexible PI substrate using screen printing and subsequently dried. (**ii**,**iii**) Using screen printing, the p-type thermoelectric ink was then applied to form a thermoelectric couple chain. Subsequently, the printed thermoelectric couple chain underwent a thermal annealing process. (**iii**–**vi**) Carbon powder was mixed into a PDMS solution to obtain a black paste, while TiO${\;\;}_{2}$ powder was mixed into a PDMS solution to obtain a white paste. These pastes were then applied by blade coating and cured to form the light-to-thermal conversion layer. Reprinted with permission from [237]. Copyright 2021, American Chemical Society. (**h**) Schematic illustration of the structure and fabrication process of PA-TENG. Reprinted with permission from [238]. Copyright 2022, Elsevier.

**Figure 22 ijms-25-06152-f022:**
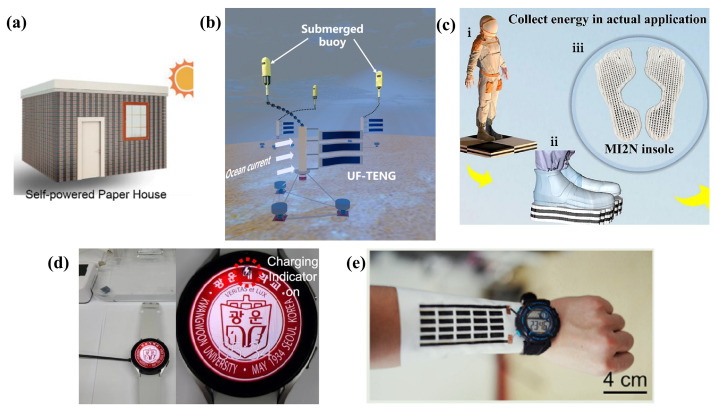
Ink-based nanogenerator applied as a power supply device. (**a**–**c**) New style ink-based nanogenerator devices. (**a**) The self-powered paper house designed to harvest energy from the hygrothermal environment. Reprinted with permission from [236]. Copyright 2023, John Wiley and Sons. (**b**) Self-powered underwater buoy for harvesting energy from low-speed ocean currents. Reprinted with permission from [38]. Copyright 2021, Elsevier. (**c**) MI2N insole for energy collection. (**i**) MI2N insole in actual application scenarios. (**ii**) Energy harvesting shoes based on MI${\;\;}_{2}$N insole. (**iii**) MI2N insole. Reprinted with permission from [46]. Copyright 2023, Elsevier. (**d**,**e**) Ink-based nanogenerators for powering electronic devices. (**d**) TENG based on MoS${\;\;}_{2}$@PVDF hybrid nanocomposite film charging for a smartwatch. Reprinted with permission from [34]. Copyright 2022, American Chemical Society. (**e**) A standard power module composed of an all-printed planar magnetoencephalography array charging for an electronic watch. Reprinted with permission from [47]. Copyright 2023, Elsevier.

**Figure 23 ijms-25-06152-f023:**
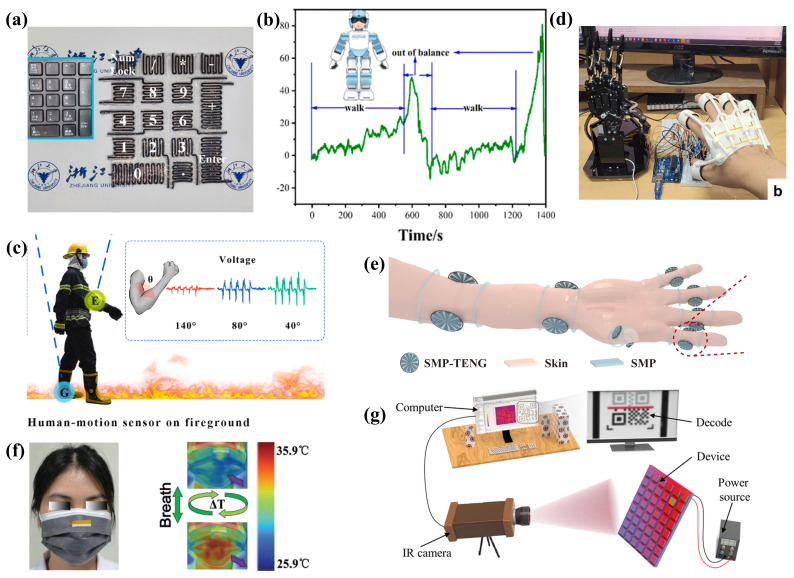
The sensing application of an ink−based nanogenerator. (**a**) Flexible self−powered keyboard. Reprinted with permission from [228]. Copyright 2021, Elsevier. (**b**) Robot gait perception. Reprinted with permission from [179]. Copyright 2023, American Chemical Society. (**c**) TENG sensors for monitoring human motion in a fireground environment. Reprinted with permission from [240]. Copyright 2021, Elsevier. (**d**) A wearable glove for controlling robotic hands in human−machine interaction. Reprinted with permission from [39]. Copyright 2023, Elsevier. (**e**) An angle sensor used for detecting human limb motion. Reprinted with permission from [182]. Copyright 2021, Elsevier. (**f**) A wearable mask capable of detecting respiratory rate. Reprinted with permission from [25]. Copyright 2023, Elsevier. (**g**) A TE infrared display with thermal imaging encryption features. Reprinted with permission from [167]. Copyright 2022, John Wiley and Sons.

**Table 1 ijms-25-06152-t001:** Overview of ink properties for ink-based TENGs.

Material	Solvent	Printing Techniques	Substrate	Post-Processing Method	Sheet Resistance	Components	Ref.
NW	carbon ink	screen printing	TPU	dried	/	electrode	[175]
CNT	FP solution	spin-coated, DIW	PEN	/	880 kΩ/sq	electrode	[176]
CNT, PEO, PEDOT:PSS	Laponite dispersion	3D printing	PDMS-PTFE	cured	/	electrode	[32]
Mxene, graphene	LiF, hydrochloric acid	inkjet printing	/	quickly dried	35 Ω/sq	electrode	[177]
Sn NPs	EtOH, EG	inkjet printing	polyimide foil	intense xenon pulsed light technology	72 ± 2 Ω/sq	electrode	[178]
BBL NPs, linear PEI	ethanol	spray-casting	/	annealed	/	dielectric layers	[33]
Ag NPs	NaCl-EG solutions	inkjet printing	fabric	ahair dryer	4.98 Ω/sqr	electrode	[179]
FKM	/	transfer printing	PI film	/	130 Ω/sqr	dielectric layers	[180]
Ag	/	inkjet printing	silicone	inert air-dried	1.52 Ω/sqr	electrode	[181]
PDMS, Ag NWs	Ag NW solution	electro hydrodynamic	AgNW	substrate heating	/	dielectric layers	[37]
SMP	ethylene glycol	FDM, spray-casting	AgNW	/	/	dielectric layers	[182]
MXene nanosheets	CNF suspension	3D printing	PET	freeze-dried	/	dielectric layers	[40]

**Table 2 ijms-25-06152-t002:** Summary of ink-based TENGs for harvesting various energy sources.

Negative PE Material	Electrode Material	Printing Method	V${\;\;}_{\mathrm{OC}}$	I${\;\;}_{\mathrm{SC}}$	Output Power	Power Density	Ref.
PTFE	C-AgNW	screen printing	12.5 V	18.4 μA	/	/	[185]
FEP	FP/CNT and ITO	DIW	/	2 mA	0.12 W	/	[176]
PDMS-PTFE	LPCC	3D-printing	27.39 V	/	/	31.64 mW m${\;\;}^{-2}$	[32]
Upilex	Sn@G	inkjet printing	150 V	5.8 mA	50 μW	125 mW m${\;\;}^{-2}$	[178]
/	AgNW	/	/	278 μA	-	/	[179]
Micropatt- erned FKM	LIG	Transfer printed	148 V	9.6 μA	280 μW	715 mW m${\;\;}^{-2}$	[180]
silicone	Ag	DIW, inkjet printing	44.16 V	/	/	1.03 W m${\;\;}^{-2}$	[181]
DMS	Copper foil, Ag NWs	electro hydrodynamic	170 V	2.52 μA	580 μW	/	[37]
SMP	AgNW	FDM, spray-casting	39 V	5.9 μA	/	56 mW m${\;\;}^{-2}$	[182]
P PTFE	Cu	coated	720 V	7.5 μA	/	/	[214]
CNF/MXene	Cu	3D printing	25 V	/	/	/	[40]
PTFE	/	screen printing	12.8 V	1.43 μA	9.1 μW	/	[38]
MXene/Ecoflex	Conductive fabric	imprint	19.91 V	/	/	/	[39]
PVDF	Al	hot microcontact printing	400 V	/	/	/	[210]
PDMS	/	3D-printing	62.3 V	/	/	/	[211]
PDMS	Al	material jet printing	76.15 V	2.07 μA	/	0.25 W m${\;\;}^{-2}$	[36]

## Data Availability

No new data were created or analyzed in this study.

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
