# Peer review of "The Latest Advances in Ink-Based Nanogenerators: From Materials to Applications"

_ijms, 2024, doi:10.3390/ijms25116152_

Round 1

Reviewer 1 Report

Comments and Suggestions for Authors

CThis review focuses on latest advances in ink-based nanogenerators. The authors analyze in detail the synthesis and doping processes of nanogenerator inks, considering different substrate materials such as metals, polymer composites and carbon-based materials and CQDs.

The review considers the challenges faced by researchers and presents the future prospects in the field.

It is notable that a substantial corpus of literature was consulted by the authors in preparing this review, with the majority of sources dating from 2020 to 2023.

The publication of this review will make a significant contribution to the field of ink materials and will be useful to researchers involved in the field of green energy production. I recommend this review for publication after minor revisions.

1. The term ink-based nanogenerators is repeated in the title and keywords. There should be no repetition.

Author Response

We greatly appreciate the time you have dedicated to providing valuable and constructive feedback. Your professional evaluation and careful review are deeply appreciated. These comments have been crucial in revising and improving our manuscript titled "The Latest Advances in Ink-Based Nanogenerators: From Materials to Applications" (ID: ijms-2979913).

The comments have been carefully considered, and thorough revisions have been made in the hope of meeting your approval. The point-by-point responses to your comments are listed in the Word document in the attachment and the revised portions are marked in red in the revised manuscript. We also provided PDF manuscript and LaTeX project files without revision marks, retaining only the revised results without the red markup.

The attachment includes the following materials:

1.Point-by-point response to your review comments (Word);

2.Revised PDF manuscript ;

3.Revised LaTeX project files;

4.Revised PDF manuscript (without revision marks);

5.Revised LaTeX project files(without revision marks) .

Reviewer 2 Report

Comments and Suggestions for Authors

This manuscript entitled “The Latest Advances in Ink-Based Nanogenerators: From Materials to Applications” is suitable for Int. J. Mol. Sci (IJMS). It explores the burgeoning field of nanogenerators, focusing on their ability to harness ambient energy from various sources such as thermal, mechanical, and moisture. It emphasizes the role of printing technologies, particularly ink-based methods, in advancing the manufacturing and structural design of nanogenerators, offering enhanced flexibility and scalability. The discussion highlights the continuous improvements in ink materials and device configurations, showcasing their potential applications in wearable electronics, motion tracking, and human-machine interactions. Overall, the manuscript provides a comprehensive framework for understanding and advancing ink-based nanogenerators, addressing both current challenges and future directions in energy harvesting technology. However, before I can accept the publication in IJMS, I think that they are several minor questions need to be asked. They are following:

1.     How do the advancements in ink-based nanogenerators, particularly in ink materials and structural design, contribute to addressing the challenges faced by the energy sector, such as the need for renewable energy sources and the development of efficient energy conversion technologies, within the context of wearable electronics and the Internet of Things (IoT)?" Your answer should access into the practical implications of the research presented, assessing its relevance to current energy challenges and emerging technologies while also considering its potential impact on future developments in the field.

2.     How do the recent advancements in ink-based thermoelectric generators (TEGs), particularly in ink materials derived from metal nanoparticles, polymer composites, carbon-based substances, and colloidal quantum dots, address the challenges associated with flexible wearable devices, such as enhancing power output, maintaining flexibility, and ensuring biocompatibility, while also considering the economic feasibility and environmental impact of utilizing rare earth elements in these ink formulations?

3.     Can you be more specific about the innovative solution-based synthesis methods for 2D transition metal dichalcogenides (TMDs), along with the development of tailored ink formulations incorporating materials like SnSe, Cu2ZnSnS4 (CZTS), and Cu2SnS3 (CTS), contribute to the advancement of solution-processed thermoelectric generators (TEGs), particularly in terms of enhancing the thermoelectric performance, optimizing charge transfer mechanisms, and enabling scalable fabrication processes?

4.     Figure 2c has lower quality and it seems that has been cut, can you replace it?

5.     How can the stability and dispersibility of solution-based 2D transition metal dichalcogenides (TMDs) inks be effectively enhanced to ensure long-term storage and optimal performance in thermoelectric generators (TEGs), considering challenges such as ink degradation, agglomeration, and precipitation during storage and printing processes?

Author Response

(The authors gave the same response as above.)

Reviewer 3 Report

Comments and Suggestions for Authors

The manuscript reports on a review of ink-based Nanogenerators, namely of triboelectric nanogenerators (TENGs), thermoelectric nanogenerators (TEGs), piezoelectric nanogenerators (PENGs), moisture-enabled nanogenerators (MEGs) and hybrid energy nanogenerators, among others. The manuscript starts by an introduction that contextualizes the overall review. Then discuss ink-based TEGs, their optimization, the materials involved, the produced systems, their geometries and fabrication processes. A similar discussion is then made for TENGs, PENGs and other generators. Subsequently, the applications of ink-based generator is discussed and the manuscript finishes with a somewhat summarized discussion on future prospects, followed by a conclusion. The discussions are done in detail about the different topics. There is an abundant amount of references related with the themes discussed in the manuscript, which help the reader to further deepen their knowledge on the subject. The figures are informative (but they need improvements) and help support the manuscript discussions. The manuscript presents an original review, but needs revisions. I have the following comments:

- The manuscript speaks of several improvements reported in the literature, but is somewhat qualitative on this. Figure-of-merit variables should have been defined and discussed in the manuscript for better comparisons. Only for TEGs the ZT figure-of-merit is mentioned. However, it is never defined in the manuscript, making it harder to read. For the other nanogenerators figures-of-merit should also be defined and presented.

- Some of the figures have small letters or numbers making them hard to read (e.g., figures 3a, 5d, 5f, 9c, etc are very hard to read even with high zoom-in ). The figures should be revised in light of this aspect.

- From time to time the words in the text seem glued together. For example, on page 7 it is written “Inadequate interfacial bondingbetween NPs leads to diminishedcarrier mobility. This is a significantbarrier to the performance of printeddevices.”. Other places also show this.

- The future prospects are very summarized. They would merit a deeper discussion in its own section, for clarity.

Author Response

(The authors gave the same response as above.)
